# Digital twin for sex-specific identification of class III antiarrhythmic drugs based on in vitro measurements, computer models, and machine learning tools

Jieyun Bai[1,2]*, Weishan Wang[1], Xiaoshen Zhang[3], Hua Lu[3]*, Henggui Zhang[4], Alexander V. Panfilov[5,6], Jichao Zhao[2]

1 Department of Cardiovascular Surgery, The first Affiliated Hospital of Jinan University, Jinan University, Guangzhou, China, 2 Auckland Bioengineering Institute, University of Auckland, Auckland, New Zealand, 3 The First Affiliated Hospital of Jinan University, Jinan University, Guangzhou, Guangzhou, China, 4 Biological Physics Group, Department of Physics and Astronomy, The University of Manchester, Manchester, United Kingdom, 5 Department of Physics and Astronomy, Ghent University, Gent, Belgium, 6 World-Class Research Center "Digital biodesign and personalized healthcare", I.M. Sechenov First Moscow State Medical University, Moscow, Russia

* jbai996@aucklanduni.ac.nz (JB); speciahual@icloud.com (HL)

## Abstract

Atrial fibrillation (AF) significantly affects morbidity and mortality rates. Class III antiarrhythmic drugs (AADs) play a crucial role in managing AF but often exhibit gender-specific complications. Our study aims to identify gender-specific Class III AADs by integrating in vitro measurements, in silico models, and machine learning (ML). By simulating drug effects on a diverse cardiomyocyte model population (5,663 males and 6,184 females), we classified drugs based on changes in action potentials and calcium transients. Using sex-dependent Support Vector Machine (SVM) algorithms, we achieved high prediction accuracy (>89%) and F1 score (>87%). Key features included changes in resting membrane potential and action potential amplitude, duration and area. Gender differences in drug responses were attributed to lower IK1, INa, and Ito in females.

## Author summary

The challenges posed by the cross-class characteristics of drugs and gender differences in drug response are significant in modern medicine. Traditional methods of drug classification fall short when dealing with drugs that act on multiple targets across different classes. Additionally, male and female patients often respond differently to the same drug due to variations in ion currents, hormonal influences, and calcium handling properties. This necessitates a shift in approach to drug classification and prescribing. In this study, we combine in vitro measurements, in silico models, and machine learning tools to address these

**Data availability statement:** The data is available at https://doi.org/10.6084/m9.figshare.28032008.v2 The code is available at https://github.com/xuanyuanzaishui/Sex-Specific-Classification-of-Antiarrhythmic-Drugs.

**Funding:** This work was supported by the Natural Science Foundation of Guangdong Province (2023A1515012833 to J.B.; 2024A1515011886 to J.B), the National Natural Science Foundation of China (61901192 to J.B.), High-end Foreign Experts Recruitment Plan of China (H20240205 to J.B.), China Scholarship Council (202206785002 to J.B.), and the Ministry of Science and Higher Education of the Russian Federation within the framework of state support for the creation and development of World-Class Research Centers "Digital Biodesign and Personalized Healthcare"(075-15-2022-304 to A.P.). The funders had no role in study design, data collection and analysis, decision to publish, or preparation of the manuscript.

**Competing interests:** The authors have declared that no competing interests exist.

challenges. We create a population of calibrated sex-specific models to simulate the effects of drugs on action potential (AP) and calcium transient (CT). By extracting biomarkers from these simulations, we train machine learning classifiers to predict true Class III antiarrhythmic drugs. Our findings show that sex-specific classifiers significantly outperform non-sex-specific classifiers in predicting drug efficacy. We identify key biomarkers that are important for differentiating male and female responses to antiarrhythmic drugs (AADs). Our study highlights the potential of computational models and machine learning in enhancing drug screening processes and developing sex-special treatment strategies. Overall, our work provides new insights into the role of gender factors in drug evaluation and paves the way for more effective and sex-special medical care.

## 1. Introduction

With a prevalence of 1–2% worldwide, atrial fibrillation (AF) is the most common cardiac rhythm disorder and a major contributor to mortality and morbidity [1–3]. Although age-adjusted AF prevalence is higher in men, women tend to have a higher risk of AF-related complications, lower quality of life and symptom management, more pronounced risk of systemic thromboembolism [4–6]. These gender disparities necessitate different management approaches, potentially leading to different outcomes. Rhythm control, promoting sinus rhythm through atrial defibrillation (cardioversion) combined with drug therapy to prevent AF recurrence, is one medical strategy [7]. However, drug therapy is thought to confer a greater risk of adverse reactions in women, and the optimal method has not been established [8–10].

Antiarrhythmic drugs (AADs) have been available for nearly 100 years and remain a mainstay in the management of AF [11,12], which is typically characterized as a reentrant arrhythmia associated with both a shortened atrial effective refractory period and reduced conduction velocity of action potentials (APs) [13]. The goals of therapy using these drugs include reducing the frequency and duration of arrhythmia episodes, as well as reducing mortality and hospitalizations associated with AF [14]. Notably, the Cardiac Arrhythmia Suppression Trial (CAST) highlighted the increased mortality risk associated with Class Ia and Class Ic AADs [15], prompting a shift towards Class III agents that prolong cardiac refractoriness. Sub-analysis of the CANT II study (Cardioversion with ANTazoline in Atrial Fibrillation II registry) illustrated that the Class Ic agent seems less efficient in men, while the Class III agent amiodarone has similar success rates in pharmacological cardioversion for both sexes [16].

Class III agents, such as sotalol, amiodarone, and dofetilide, exert their primary antiarrhythmic effects by blocking the rapid component of the delayed rectifier outward potassium current (IKr). This action extends the effective refractory period of atrial myocardial cells, thereby prolonging phases 2 and 3 of APs. This pharmacological mechanism enables reentrant impulses to encounter refractory tissue, facilitating the termination of AF and the restoration of sinus rhythm [17]. While agents are categorized in Vaughan Williams Class III [18], they exhibit characteristics spanning

multiple classes. For example, amiodarone, the archetype of this class, acts on targets spanning all four Vaughan Williams classes. Other Class III agents also exhibit additional properties; sotalol, for instance, functions as a beta-blocker, and ibutilide affects slow inward depolarizing sodium currents. Consequently, a "pure" potassium channel blocker effect is challenging to exemplify. Although Class III agents can block potassium channels, many non-Class III also impact these channels. Current guidelines necessitate the measurement of AP duration (APD) and QT interval prolongation to classify new drugs as Class III or assess their torsade de pointes (TdP) risk [19,20]. These guidelines have effectively identified Class III drugs and prevented cardiotoxic drugs from reaching the market. However, they also present limitations as biomarkers and may introduce bias in therapeutic decisions and the development of novel AADs.

In response to these limitations, a paradigm has emerged, integrating in vitro studies to assess drugs' effects on various ion channels with in silico models of cardiac myocyte electrophysiology [21–26]. This approach enhances our understanding of how these effects collectively influence cardiac function, providing a robust tool for drug screening [27–34]. Mathematical models of cardiac electrophysiology allow for the simulation of various conditions [35–39], offering insights typically obtained through animal experiments [40,41]. Computational methods have thus become essential in evaluating drugs' effects, with simulated measurements from biophysical models being applicable to machine learning (ML) pipelines [42–45]. These pipelines can uncover mechanistic insights often overlooked in conventional analyses. However, to our knowledge, no simulation-based approach has incorporated gender as a variable in evaluating AADs for AF [46]. It is well-established that AADs may cause more complications in women than in men [47–51], suggesting the need for sex-specific classifiers [52–54]. Despite this, female representation is significantly lacking in both basic research and clinical studies related to drug development [55,56]. In vitro studies predominantly use male animals, raising concerns about the generalizability of findings. This gender bias extends to mathematical models of cardiac cells, which are often parameterized using male-dominated datasets. The underrepresentation of women in clinical cohorts further exacerbates the challenge, as it hinders the training of accurate classifiers due to a lack of reliable ground truth data. Addressing this gender disparity is crucial for improving the safety and efficacy of AADs for all patients [57].

Main aim of this study is to develop a pipeline which can be used for in-silico sex-specific drug testing. Specifically, we are interested in class III drugs effective for patients with AF and account for differences between male and female patients. This pipeline is based on a recent study [58] where multivariable analysis was performed of individual action potential (AP) data from trabeculae obtained during heart surgery of patients in longstanding AF (n = 201) and seven AP biomarkers were extracted for male (n = 126) and female (n = 75) patients. Based on those biomarkers, we develop sex specific population models. Using it we simulated the effects of 12 drugs clinically used (i.e., 6 Class III and 6 non-Class III AADs) under AF conditions. We then fed the resulting datasets of simulated biomarkers which include 7 AP biomarkers [58] and 7 additional biomarkers to ML algorithms to generate male and female classifiers to identify Class III AADs. Finally, we used five-fold cross-validation to evaluate and validate the performance of sex-specific models (Fig 1). We describe in details the results and conclude that these models provide powerful tools for drug screening and personalized medicine. Also, our findings identify key features that need to be considered in drug classification in relation to gender factors.

## 2. Results

### 2.1. Construction and calibration of sex-special population of atrial fibrillation model

We generated a large initial population of 100,000 models by randomly varying key parameters related to human atrial electrophysiology. This population was calibrated to retain only those models that were fully consistent with the experimentally observed ranges of seven biomarkers that inherently capture the integrated effects of diverse clinical factors (e.g., age, sex, body mass index, hypertension, left ventricular hypertrophy, AF and beta-blocker usage) on human atrial electrophysiology [58]. The calibration process reduced the initial population to 11,847 accepted models (5,663 males and 6,184 females) (See Method 5.3).

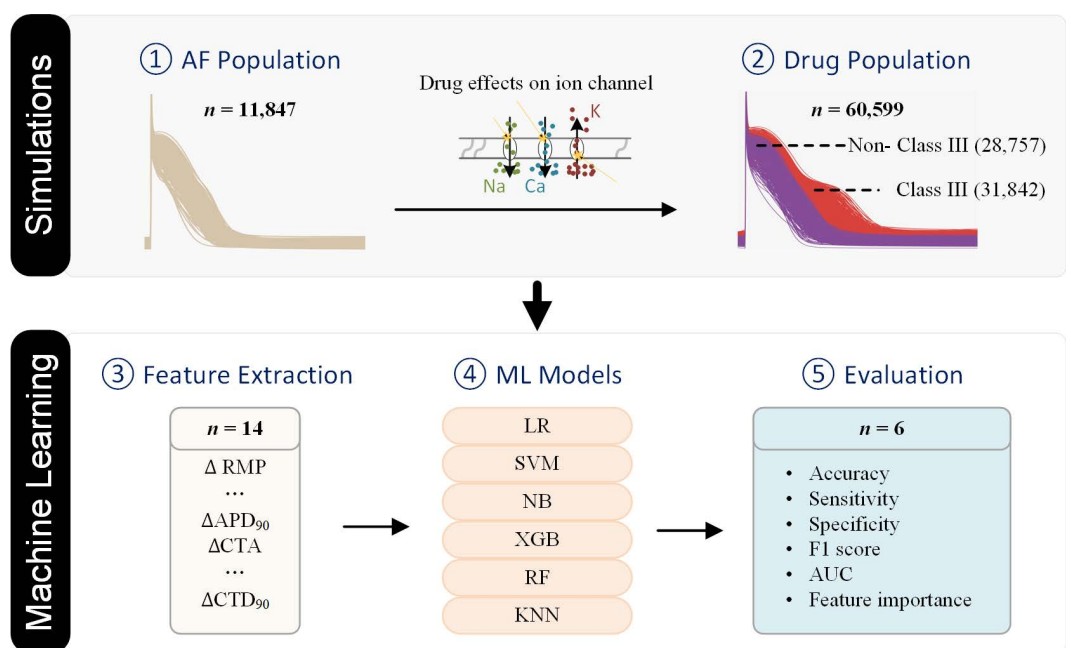

**Fig 1. The classification framework with cardiac simulations and machine learning (ML) models.** ① An atrial fibrillation (AF) population of cell models was constructed by generating various combinations of ionic currents based on ex vivo experimental data; ② A drug population was created by incorporating the blocking effects of Class III and Non-Class III antiarrhythmic drugs on ion channels into the AF population; ③ Fourteen biomarkers were extracted from the action potential (AP) and calcium transient (CT) of each sample to form a dataset for drug classification. ④ Six ML models were trained and validated using the 5-fold cross-validation technique; and ⑤ The classification performance was evaluated using six metrics. RMP - resting membrane potential; APD90 - AP duration at 90% of repolarization; CTA - CT amplitude; CTD90 - CT duration at 90% of repolarization; LR - Logistic Regression; SVM - Support Vector Machine; NB - Naive Bayes; XGB - Extreme Gradient Boosting; RF - Random Forest; KNN - K-Nearest Neighbors; and AUC - Area Under the Curve of Receiver Operating Characteristic.

Fig 2 illustrates the time course of different APs and CTs accepted into the population, along with the corresponding distributions of fourteen biomarkers, considered in experimental study [47] including APD at 20%, 40%, 50%, and 90% of repolarization (APD20, APD40, APD50, and APD90, respectively), AP amplitude (APA), resting membrane potential (RMP), and maximum upstroke velocity (dV/dt_max), APD triangulation (APD_tri, defined as APD90 – APD50). We also added to our analysis biomarkers based on calcium dynamics of our models, which was not possible to obtain in experiments. In particular, we considered diastolic calcium concentration (CTD), the maximum of CT (CT_max), CT amplitude (CTA), CT duration from peak to 50% and 90% repolarization (CTD50 and CTD90, respectively), and CT triangulation (CTD_tri, defined as CTD90 – CTD50). Sex differences are observed in AP recordings, showing less negative RMP, smaller dV/dt_max and APA, and larger APDs in female AP recordings, with no visible sex differences in CT biomarkers.

### 2.2. Variable responses to class III and non-class III drugs across the atrial fibrillation population

We examined the electrophysiological responses of human atrial myocyte models to Class III and Non-Class III drugs. By incorporating drug effects on different ion channels, we created an initial population from the AF population and removed models with abnormalities to generate a drug population of 60,599 AP models (See Method 5.4). Fig 3 displays the time course of various APs and CTs and the distributions of biomarkers in the Class III ($n = 31,842$) and non-Class III ($n = 28,757$) subpopulations. The Class III subpopulation exhibited longer APs and higher CTs, more negative RMP, larger dV/dt_max and APA, and longer APD, with statistical significance ($p < 0.001$).

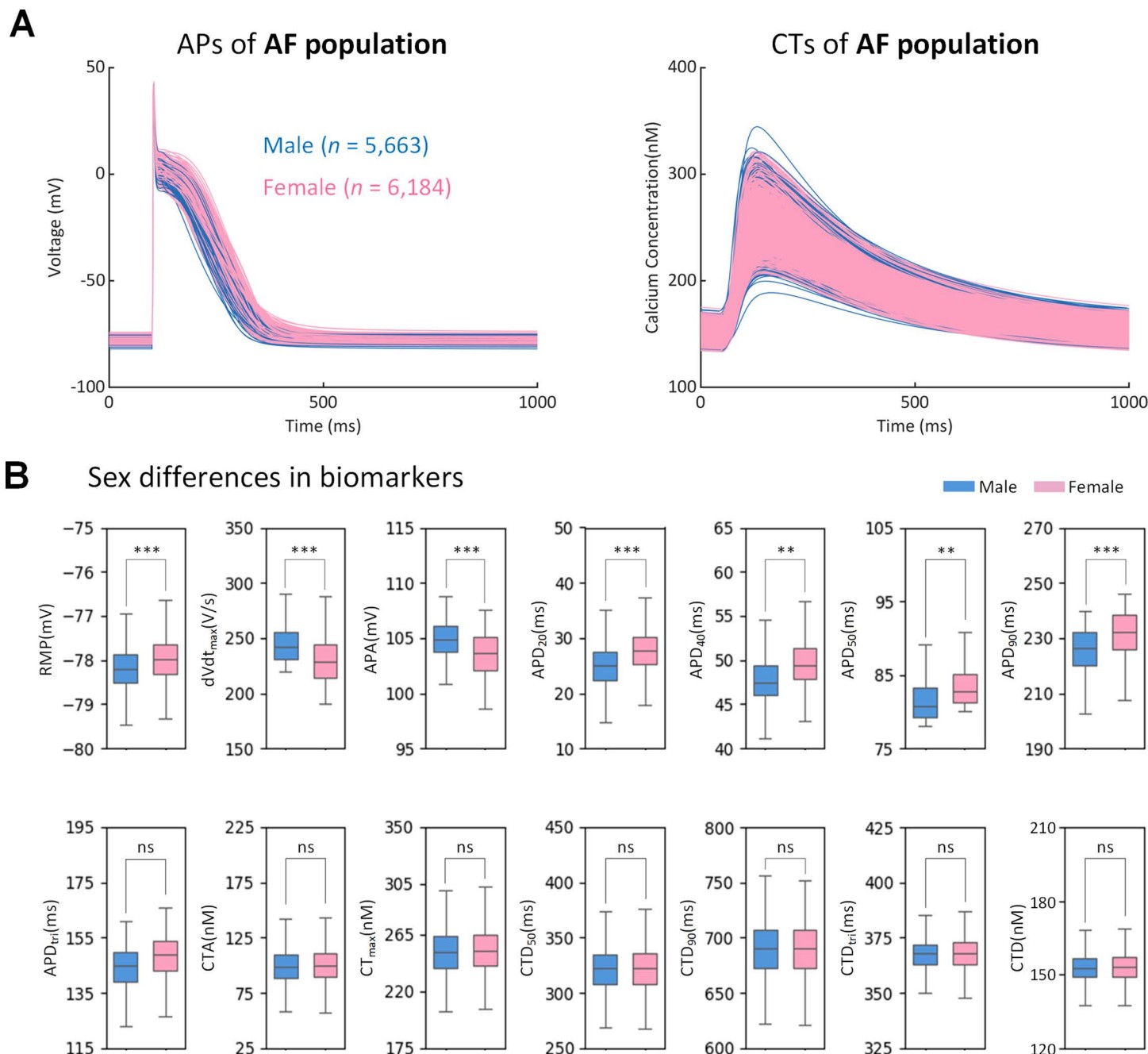

**Fig 2. Sex differences in the AF population of human atrial cell models (5,663 males and 6,184 females).** (A) Traces of action potential (AP) and calcium transient (CT). (B) Sex differences in AP biomarkers (RMP, dV/dt$_{max}$, APA, APD$_{20}$, APD$_{40}$, APD$_{50}$, APD$_{90}$ and APD$_{tri}$) (CTA, CT$_{max}$, CTD$_{50}$, CTD$_{90}$, CTD$_{tri}$ and CTD). The p-values are indicated and the stars *, **, *** indicate a p-value < 0.05, 0.01 and 0.001, respectively. ns indicates not significant.

Fig 4 shows male (n = 28,967) and female (n = 31,632) AP models in the drug population. Compared to the male subpopulation, the female subpopulation displayed less negative RMP, smaller dV/dt_max and APA, longer APDs and APD_tri, larger CTA, CT_max and CTD, and smaller CTD50, CTD90 and CTD_tri, with statistical significance (p < 0.001).

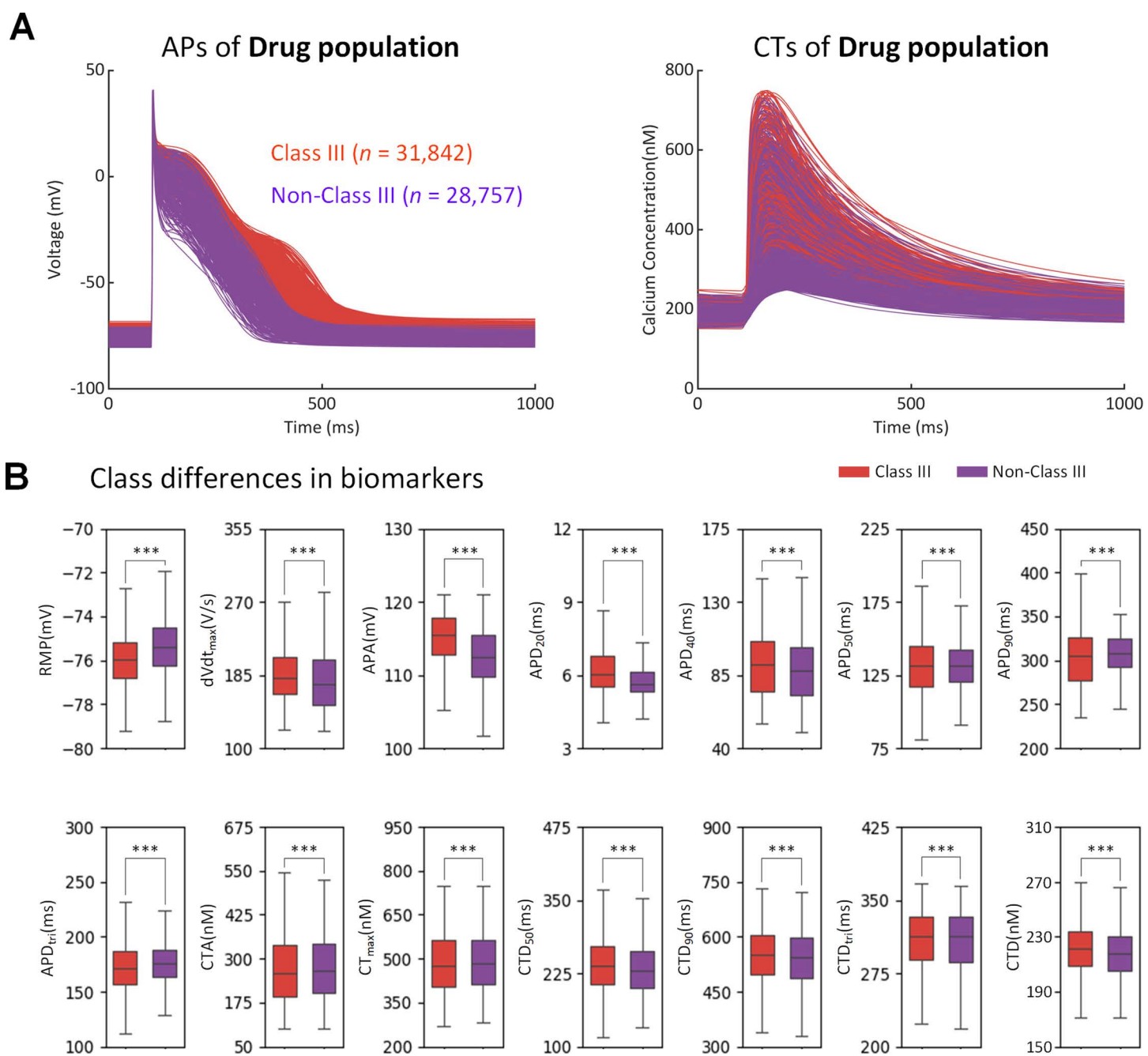

**Fig 3. Class differences in the drug population of human atrial cell models (31,842 in the Class III and 28,757 in the non-Class III).** (A) Traces of action potential (AP) and calcium transient (CT). (B) Class differences in AP biomarkers and CT biomarkers. *** indicates a p-value < 0.001.

### 2.3. Changes in biomarkers before and after medication predict the drug class

After establishing populations before (i.e., AF) and after medication (i.e., Class III or Non-Class III drugs), we extracted features to train ML models for drug classification by quantifying changes in biomarkers. The goal of this analysis was to identify other features besides ΔAPD for representing the drug's efficacy. Fig 5 shows consistent feature distributions

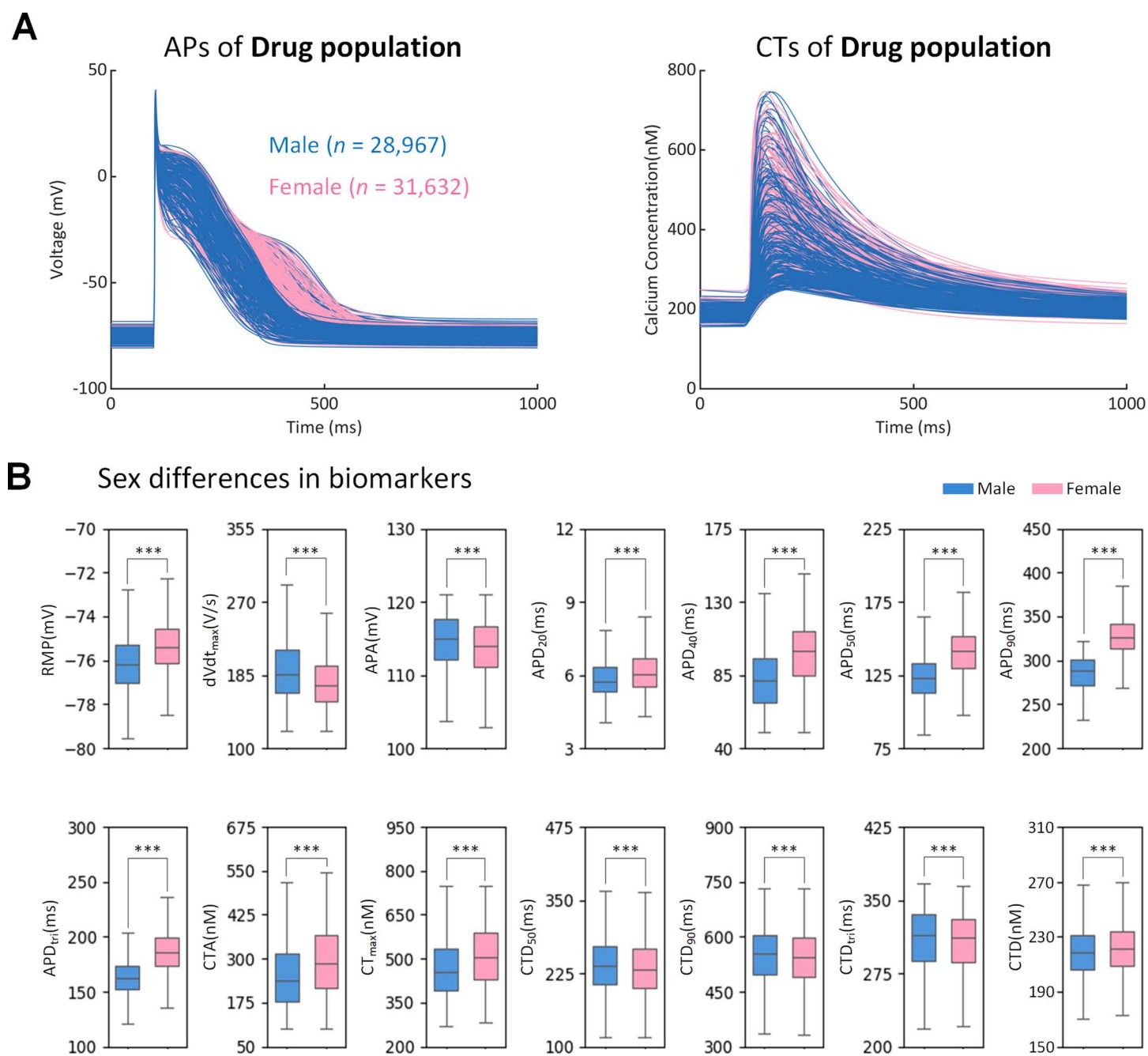

**Fig 4. Sex differences in the drug population of human atrial cell models (28,967 males and 31,632 females).** (A) Traces of action potential (AP) and calcium transient (CT). (B) Sex differences in AP biomarkers and CT biomarkers. *** indicates a p-value < 0.001.

between training and testing data sets but significant differences between Class III and non-Class III subpopulations, mainly in AP features such as ΔRMP, ΔdV/dtmax and ΔAPA, ΔAPD20, ΔAPD40, ΔAPD50, and ΔAPD90.

Using these features, we trained and validated six ML models, then evaluated their performance on the testing data set. Six classic ML models (LR, SVM, NB, XGB, RF, KNN) were chosen for their established effectiveness and wide

## A  Features of action potential (AP)

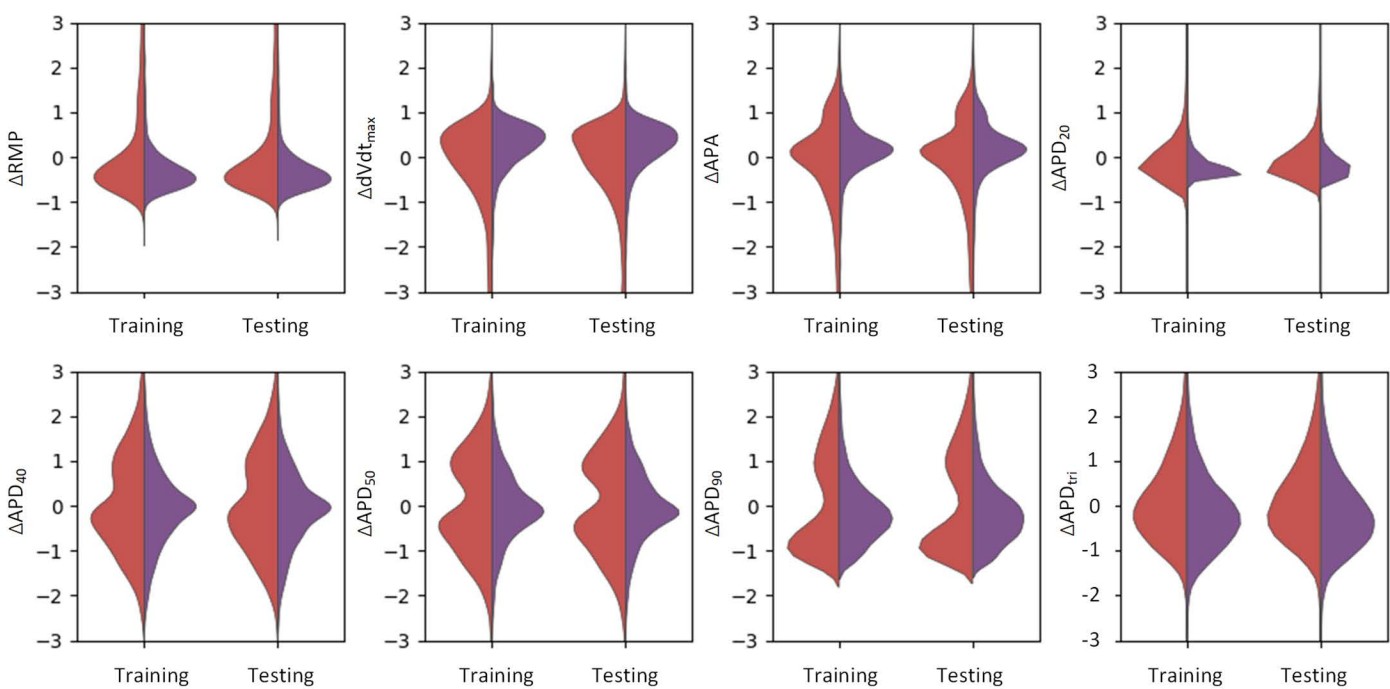

## B  Features of calcium transient (CT)

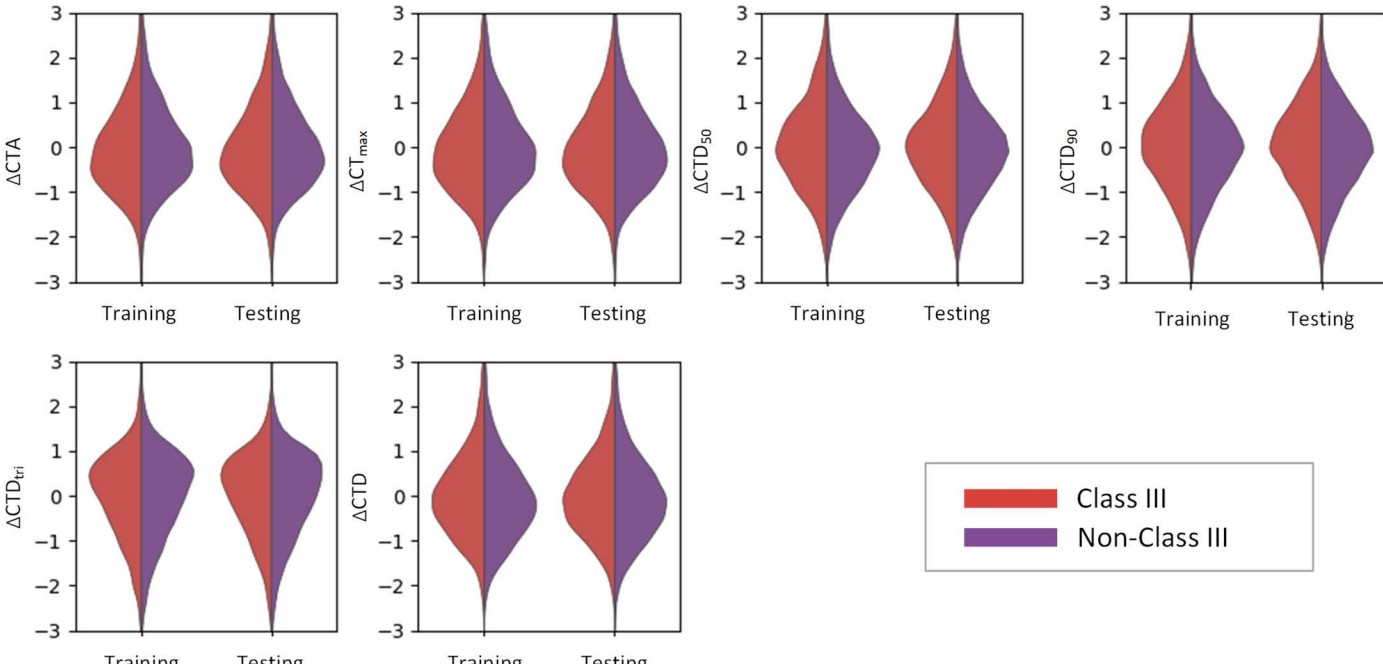

**Fig 5. The feature distributions in the training and testing sets are similar, but the feature distributions in the Class III and non-Class III categories are partly different.** (A) Action potential features ($\Delta$RMP, $\Delta$dV/dt$_{max}$, $\Delta$APA, $\Delta$APD$_{20}$, $\Delta$APD$_{40}$, $\Delta$APD$_{50}$, $\Delta$APD$_{90}$ and $\Delta$APD$_{tri}$). (B) Calcium transient features ($\Delta$CTA, $\Delta$CT$_{max}$, $\Delta$CTD$_{50}$, $\Delta$CTD$_{90}$, $\Delta$CTD$_{tri}$ and $\Delta$CTD).

usage in various classification problems [52], making them reliable choices for our task. We found that the SVM classifier outperformed others with an accuracy (ACC) of 0.8095 [95% CI, 0.8064-0.8127], a specificity (SPE) of 0.8217 [95% CI, 0.8177-0.8257], an AUC of 0.8085 [95% CI, 0.8053 - 0.8117] and F1 score of 0.8226 [95% CI, 0.8193 - 0.8259]. Also, XGB classifier showed comparable performance, followed by KNN (Table 1 and Fig 6), with statistical significance (p<0.05).

## 2.4. Considering sex differences improve drug classification performance

We aimed to determine if classification performance improves by considering sex differences. For that we developed sex-specific classifiers by training and testing each of 6 classifiers on data sets from virtual patients of the same gender. Fig 7 shows that the best male classifier is SVM which achieved an AUC of 0.8669 [95% CI, 0.8583 - 0.8707] and an F1

**Table 1. Performance of sex-specific and non-sex-specific classifiers.**

| Type | Models | ACC | SEN | SPE | F1 | MCC | AUC |
|---|---|---|---|---|---|---|---|
| Sex-special model (Male) | LR | 0.6553 (0.6460,0.6646) | 0.6063 (0.5952,0.6174) | 0.6198 (0.6046,0.6351) | 0.6122 (0.6071,0.6173) | 0.3025 (0.2820,0.3229) | 0.6507 (0.6428,0.6584) |
| | SVM | **0.8802 (0.8751,0.8853)** | 0.7635 (0.7569,0.7701) | **0.9454 (0.9320,0.9587)** | **0.8450 (0.8389,0.8510)** | **0.7620 (0.7478,0.7762)** | **0.8659 (0.8607,0.8703)** |
| | NB | 0.5993 (0.5949,0.6036) | **0.9346 (0.9266,0.9427)** | 0.5166 (0.5137,0.5195) | 0.6652 (0.6617,0.6687) | 0.3350 (0.3227,0.3474) | 0.6422 (0.6380,0.6463) |
| | XGB | 0.8696 (0.8614,0.8777) | 0.7826 (0.7707,0.7944) | 0.8975 (0.8849,0.9100) | 0.8368 (0.8267,0.8469) | 0.7129 (0.7000,0.7259) | 0.8586 (0.8503,0.8669) |
| | RF | 0.8471 (0.8410,0.8531) | 0.7550 (0.7459,0.7641) | 0.8702 (0.8563,0.8841) | 0.8084 (0.8014,0.8154) | 0.6929 (0.6767,0.7091) | 0.8354 (0.8296,0.8413) |
| | KNN | 0.8263 (0.8228,0.8297) | 0.7010 (0.69430.7077) | 0.8663 (0.8554,0.8772) | 0.7752 (0.7716,0.7788) | 0.6461 (0.6360,0.6561) | 0.8103 (0.8073,0.8133) |
| Sex-special model (Female) | LR | 0.6707 (0.6666,0.6747) | 0.5790 (0.5760,0.5820) | 0.6225 (0.6166,0.6284) | 0.5999 (0.5961,0.6038) | 0.3210 (0.3108,0.3312) | 0.6589 (0.6551,0.6627) |
| | SVM | **0.8774 (0.8738,0.8809)** | 0.7660 (0.7613,0.7707) | **0.9484 (0.9442,0.9525)** | **0.8475 (0.8429,0.8520)** | **0.7601 (0.7509,0.7691)** | **0.8663 (0.8626,0.8699)** |
| | NB | 0.5877 (0.5833,0.5921) | **0.9183 (0.9135,0.9232)** | 0.5207 (0.5177,0.5237) | 0.6647 (0.6627,0.6666) | 0.2918 (0.2839,0.2996) | 0.6206 (0.6170,0.6242) |
| | XGB | 0.8579 (0.8540,0.8619) | 0.7788 (0.7726,0.7850) | 0.8894 (0.8823,0.8965) | 0.8295 (0.8249,0.8342) | 0.6896 (0.6765,0.7026) | 0.8499 (0.8460,0.8538) |
| | RF | 0.8320 (0.8292,0.8347) | 0.7476 (0.7421,0.7531) | 0.8571 (0.8521,0.8620) | 0.7980 (0.7946,0.8014) | 0.6720 (0.6617,0.6823) | 0.8235 (0.8206,0.8263) |
| | KNN | 0.8201 (0.8159,0.8242) | 0.7008 (0.6951,0.7065) | 0.8704 (0.8621,0.8787) | 0.7757 (0.7708,0.7806) | 0.6396 (0.6286,0.6507) | 0.8081 (0.8040,0.8122) |
| Non-sex-special model | LR | 0.6717 (0.6696,0.6737) | 0.8385 (0.8342,0.8427) | 0.6501 (0.6478,0.6524) | 0.7323 (0.7309,0.7338) | 0.3430 (0.3389,0.3471) | 0.6589 (0.6566,0.6611) |
| | SVM | **0.8095 (0.8064,0.8127)** | 0.8253 (0.8187,0.8319) | **0.8217 (0.8177,0.8257)** | **0.8226 (0.8193,0.8259)** | **0.6179 (0.6101,0.6258)** | **0.8085 (0.8053,0.8117)** |
| | NB | 0.6325 (0.6279,0.6371) | **0.9012 (0.8958,0.9066)** | 0.6055 (0.6025,0.6084) | 0.7243 (0.7208,0.7278) | 0.2776 (0.2660,0.2892) | 0.6118 (0.6072,0.6165) |
| | XGB | 0.7914 (0.7879,0.7949) | 0.7959 (0.7868,0.8050) | 0.8112 (0.8079,0.8145) | 0.8034 (0.7991,0.8077) | 0.5815 (0.5748,0.5881) | 0.7910 (0.7878,0.7943) |
| | RF | 0.7650 (0.7615,0.7685) | 0.7858 (0.7793,0.7923) | 0.7778 (0.7744,0.7813) | 0.7818 (0.7781,0.7854) | 0.5264 (0.5220,0.5309) | 0.7634 (0.7600,0.7668) |
| | KNN | 0.7844 (0.7801,0.7886) | 0.8014 (0.7959,0.8068) | 0.7972 (0.7922,0.8023) | 0.7993 (0.7954,0.8032) | 0.5665 (0.5578,0.5751) | 0.7831 (0.7787,0.7874) |

## Non-sex-specific classifiers

**Fig 6. Prediction performances of non-sex-specific classifiers.** Receiver operating characteristic curves (left) and F1 scores (right) are shown. *p<0.05.

score of 0.8482 [95% CI, 0.8376 - 0.8530]. SVM is also the best female classifier achieved an AUC of 0.8659 [95% CI, 0.8554 - 0.8726] and an F1 score of 0.8456 [95% CI, 0.8324 - 0.8540]. These classifiers outperformed the non-sex-specific classifier, increasing the ACC for SVM by ~7%, the SPE by 12%, the AUC by ~6% and the F1 score by ~2%. Close performance was shown by XGB classifier (Table 1).

We examined if classification performance decreases by developing sex-biased classifiers, trained and tested on data sets from virtual patients of the different gender. As is shown in the Fig 8, the SVM trained on a male dataset set but tested on a female dataset achieved an AUC of 0.8363 [95% CI, 0.8355-0.8372] and an F1 score of 0.8101 [95% CI, 0.8090-0.8112], while it trained on a female dataset but tested on a male dataset achieved an AUC of 0.8571 [95% CI, 0.8559-0.8582] and an F1 score of 0.8353 [95% CI, 0.8337-0.8369]. These sex-biased classifiers are inferior to sex-specific classifiers, reducing the ACC by ~3%, the SPE by ~1% and the F1 score by ~1% (Table 2). These data indicate the importance of considering sex differences in predicting drug efficacy.

Further experiments were conducted to investigate the impact of the number of features on classification performance. When using different numbers (i.e., 3, 5, 7, 9, 11 or 14) of top features, both of sex-specific classifiers show that all 14 features are essential for achieving optimal performance. Important to note that APD features alone obtained in experiment are insufficient to predict drug efficacy and one needs to use CT features which we introduced in our study (Table 3).

We also conducted a dimension reduction analysis such as principal component analysis (PCA). The Male model with 12 principal components (AUC: 0.8661 vs. 0.8655) and the Female model with 11 principal components (AUC: 0.8687vs. 0.8663) achieves better performance (See Table 4). These results showed that indeed a reduced set of features could potentially achieve better performance.

### 2.5. The different roles of ΔRMP and ΔAPA in sex-special classifiers

To analyze differences in sex-specific classifiers, we performed feature importance analysis using SHAP values. Fig 9 shows feature importance rankings in sex-specific classifiers. Both male and female classifiers found ΔAPD90 as the most important feature, with AP features (except for APD40) generally more important than CT features. However, excluding CT features substantially decreases the accuracy of the prediction (Table 3). The main difference between

## A    Sex-specific classifiers (**Male**)

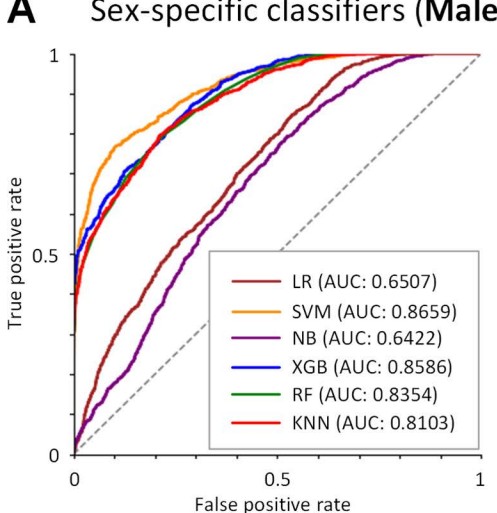
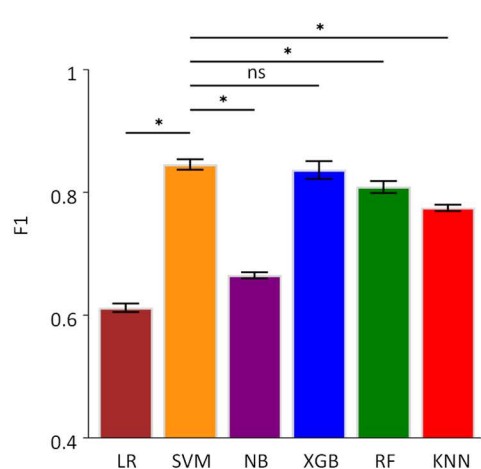

## B    Sex-specific classifiers (**Female**)

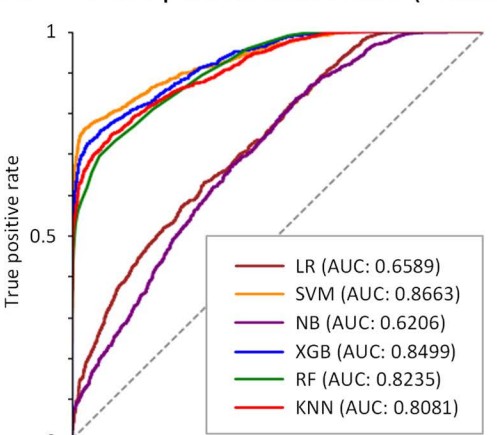
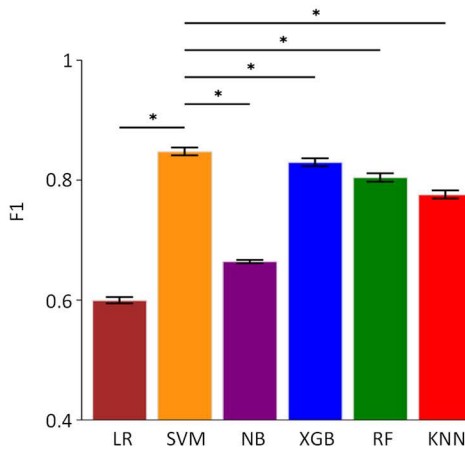

**Fig 7. Prediction performances of sex-specific classifiers.** Receiver operating characteristic curves and F1 scores of male (A) and female classifiers (B) are shown. *p < 0.05 and ns indicates not significant.

Male and Female classifiers was in the ranking of ΔRMP and ΔAPA. Compared with feature importance ranking for the male classifier (Fig 9A), ΔRMP is less important than ΔAPA in the feature importance ranking for the female classifier (Fig 9B). Further correlation analysis showed a high negative correlation between ΔRMP and ΔAPA (correlation coefficient of -0.88) (Fig 9C).

We also conducted ablation experiments. In these experiments, we removed the feature ΔRMP or ΔAPA to illustrate the impact of these features on classification performance (See Table 5). The experimental results show that ΔRMP and ΔAPA have different importance in male and female classifiers. In male classifiers, ΔRMP plays a greater role than ΔAPA, while ΔRMP plays a smaller role than ΔAPA in the female classifier. This provides additional evidence to support our claim about the differences in the ranking of these features between male and female classifiers (See Table 6).

## A    Sex-biased classifiers (**Female on Male**)

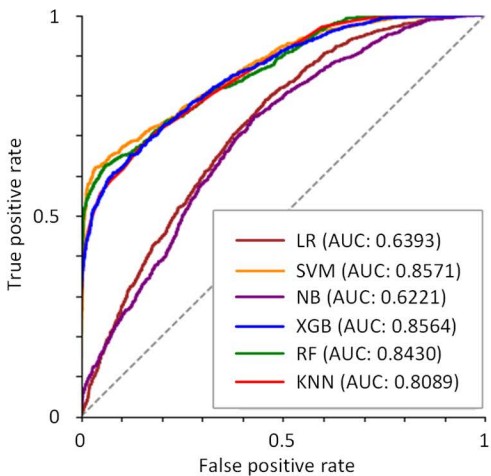

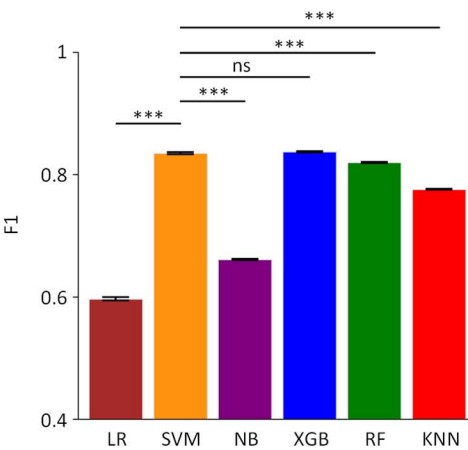

## B    Sex-biased classifiers (**Male on Female**)

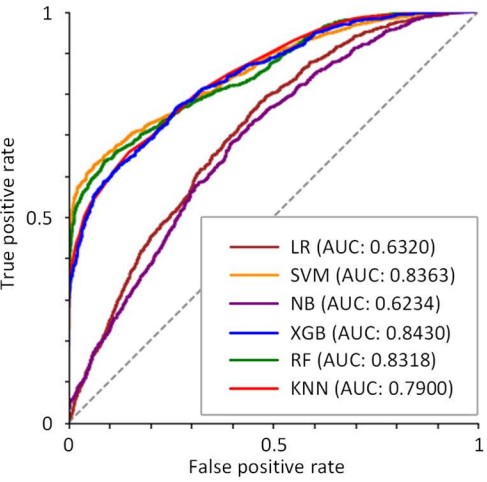

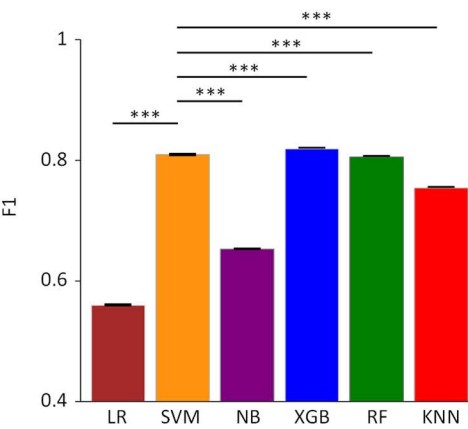

**Fig 8. Prediction performances of sex-biased classifiers.** Female on Male, models are trained on a female dataset but tested on a male dataset, but Male on Female, models are trained on a male dataset but tested on a female dataset. ***$p < 0.001$, and ns, not significant.

### 2.6. Effects of the new geometric biomarker on classification performance

Feature importance analysis revealed that ΔRMP, ΔAPA, and ΔAPD90 were the three most critical features. Interestingly, the combination of RMP, APA, and APD90 effectively represents the area under the action potential curve (AREA). Therefore, we extracted this composite feature (ΔAREA) and investigated its effects on classification performance and its role in the predictive model. Experimental results demonstrated that incorporating this new biomarker led to a consistent improvement by ~2% in the model's AUC (See Table 7). Furthermore, subsequent feature importance analysis ranked this newly derived biomarker as the fourth most influential in the classification process (Fig A in S1 Text).

**Table 2. Performance of sex-biased classifiers (i.e., models trained and tested on the data set with virtual patients of the different gender). M-male and F-female.**

| Type | Models | ACC | SEN | SPE | F1 | MCC | AUC |
|---|---|---|---|---|---|---|---|
| **Male on Female** | LR | 0.6473 (0.6470,0.6476) | 0.5248 (0.5228,0.5268) | 0.6018 (0.6012,0.6024) | 0.5607 (0.5596,0.5617) | 0.2702 (0.2694,0.2709) | 0.6320 (0.6316,0.6325) |
| | SVM | 0.8472 (0.8465,0.8479) | 0.7601 (0.7580,0.7622) | **0.8671 (0.8665,0.8678)** | 0.8101 (0.8090,0.8112) | 0.6874 (0.6860,0.6889) | 0.8363 (0.8355,0.8372) |
| | NB | 0.5808 (0.5800,0.5817) | **0.9227 (0.9197,0.9256)** | 0.5061 (0.5056,0.5067) | 0.6537 (0.6534,0.6540) | 0.2959 (0.2950,0.2967) | 0.6234 (0.6230,0.6238) |
| | XGB | **0.8492 (0.8481,0.8502)** | 0.7997 (0.7963,0.8031) | 0.8408 (0.8375,0.8441) | **0.8197 (0.8186,0.8209)** | **0.6909 (0.6889,0.6930)** | **0.8430 (0.8420,0.8440)** |
| | RF | 0.8396 (0.8389,0.8402) | 0.7775 (0.7741,0.7809) | 0.8368 (0.8357,0.8379) | 0.8060 (0.8047,0.8074) | 0.6709 (0.6695,0.6723) | 0.8318 (0.8308,0.8328) |
| | KNN | 0.8005 (0.7988,0.8021) | 0.7166 (0.7143,0.7190) | 0.7976 (0.7927,0.8025) | 0.7549 (0.7539,0.7560) | 0.5900 (0.5865,0.5934) | 0.7900 (0.7888,0.7912) |
| **Female on Male** | LR | 0.6446 (0.6430,0.6462) | 0.5872 (0.5823,0.5921) | 0.6079 (0.6055,0.6104) | 0.5974 (0.5951,0.5997) | 0.2797 (0.2765,0.2827) | 0.6393 (0.6377,0.6408) |
| | SVM | **0.8695 (0.8685,0.8704)** | 0.7427 (0.7394,0.7459) | **0.9545 (0.9530,0.9560)** | 0.8353 (0.8337,0.8369) | **0.7458 (0.7442,0.7475)** | **0.8571 (0.8559,0.8582)** |
| | NB | 0.5958 (0.5950,0.5966) | **0.8801 (0.8778,0.8824)** | 0.5301 (0.5295,0.5306) | 0.6617 (0.6609,0.6624) | 0.2790 (0.2771,0.2809) | 0.6221 (0.6213,0.6229) |
| | XGB | 0.8638 (0.8628,0.8649) | 0.7882 (0.7853,0.7910) | 0.8939 (0.8896,0.8983) | **0.8377 (0.8368,0.8386)** | 0.7256 (0.7231,0.7280) | 0.8564 (0.8556,0.8573) |
| | RF | 0.8530 (0.8522,0.8537) | 0.7458 (0.7437,0.7479) | 0.9106 (0.9077,0.9135) | 0.8200 (0.8191,0.8209) | 0.7078 (0.7059,0.7096) | 0.8430 (0.8423,0.8438) |
| | KNN | 0.8206 (0.8197,0.8215) | 0.6936 (0.6898,0.6973) | 0.8817 (0.8765,0.8870) | 0.7764 (0.7756,0.7772) | 0.6429 (0.6404,0.6453) | 0.8089 (0.8082,0.8095) |

## 2.7. Lower levels of IK1, INa and Ito in females versus males explain the gender difference in drug response

To determine the ionic mechanisms underlying AP differences between male and female AF patients, we identified key ion currents related to RMP and APA via parameter sensitivity analysis and examined statistical differences between male and female AF models. Fig 10A shows that RMP is sensitive to GK1 and GNaK, while APA is sensitive to GK1, GNaK, GNa, and Gto. However, there is no sex difference in GNaK but in GK1, GNa, and Gto, with these currents being smaller in females. In addition, there are sex differences in GKs, GKur, GCaL, GNaCa, Grel and Gleak (Fig 10B).

## 3. Discussion

In this study, we developed a computational pipeline that combines mechanistic modeling with ML analyses, and we applied this to examine individualized drug response. Mechanistic simulations were used to generate an experimentally-calibrated population of virtual atrial cells representing AF patients and the corresponding drug population of thousands of cardiomyocytes representing AF patients after taking medication. Several ML algorithms were applied to detect how the members of the AF population would respond to the Class III and non-Class III drugs. Importantly, ML was not employed to examine the arrhythmic dynamics, a task that can often be performed by visual inspection, but to detect how AADs affect physiological waveforms under AF condition. The ML analyses indicated that the gender factor needs to be considered in drug efficacy assessments. Our main findings included: (1) Significant differences in AP and CT biomarkers between virtual male and female patients within the drug population; (2) AP biomarkers were more important than CT biomarkers in drug classification, though CT biomarkers were essential for optimal classification accuracy; (3) ΔAPA and ΔRMP were identified as key features reflecting differences between male- and female-specific classifiers;

**Table 3. Performance of SVM using different numbers of top biomarkers based on feature importance ranking shown in Fig 9.**

| Type | Number | ACC | SEN | SPE | F1 | MCC | AUC |
|---|---|---|---|---|---|---|---|
| **Sex-special model (Male)** | 3 | 0.7519 (0.7461,0.7577) | 0.6406 (0.6306,0.6506) | 0.7424 (0.7345,0.7504) | 0.6877 (0.6798,0.6957) | 0.4876 (0.4753,0.4998) | 0.7377 (0.7315,0.7438) |
| | 5 | 0.8021 (0.7964,0.8079) | 0.6781 (0.6677,0.6885) | 0.8269 (0.8177,0.8362) | 0.7451 (0.7371,0.7530) | 0.5938 (0.5817,0.6060) | 0.7862 (0.7802,0.7923) |
| | 7 | 0.8590 (0.8543,0.8637) | 0.7285 (0.7179,0.7391) | 0.9250 (0.9192,0.9309) | 0.8150 (0.8079,0.8221) | 0.7169 (0.7075,0.7263) | 0.8423 (0.8369,0.8476) |
| | 9 | 0.8694 (0.8612,0.8776) | 0.7446 (0.7339,0.7552) | 0.9364 (0.9198,0.9530) | 0.8294 (0.8191,0.8398) | 0.7385 (0.7206,0.7565) | 0.8534 (0.8452,0.8616) |
| | 11 | 0.8738 (0.8674,0.8802) | 0.7530 (0.7414,0.7646) | 0.9391 (0.9265,0.9517) | 0.8357 (0.8272,0.8443) | 0.7472 (0.7336,0.7608) | 0.8583 (0.8516,0.8650) |
| | **14** | **0.8802 (0.8751,0.8853)** | **0.7635 (0.7569,0.7701)** | **0.9454 (0.9320,0.9587)** | **0.8450 (0.8389,0.8510)** | **0.7620 (0.7478,0.7762)** | **0.8655 (0.8607,0.8703)** |
| **Sex-special model (Female)** | 3 | 0.7299 (0.7254,0.7344) | 0.6576 (0.6513,0.6638) | 0.7129 (0.7066,0.7192) | 0.6841 (0.6789,0.6893) | 0.4501 (0.4409,0.4592) | 0.7227 (0.7182,0.7272) |
| | 5 | 0.7843 (0.7783,0.7902) | 0.7116 (0.7027,0.7204) | 0.7834 (0.7771,0.7898) | 0.7458 (0.7382,0.7533) | 0.5612 (0.5490,0.5735) | 0.7770 (0.7708,0.7833) |
| | 7 | 0.8511 (0.8464,0.8558) | 0.7395 (0.7301,0.7489) | 0.9088 (0.9020,0.9155) | 0.8154 (0.8090,0.8219) | 0.7033 (0.6938,0.7128) | 0.8400 (0.8350,0.8450) |
| | 9 | 0.8635 (0.8578,0.8692) | 0.7460 (0.7388,0.7532) | 0.9338 (0.9231,0.9446) | 0.8294 (0.8224,0.8363) | 0.7306 (0.7184,0.7429) | 0.8518 (0.8461,0.8575) |
| | 11 | 0.8703 (0.8656,0.8750) | 0.7569 (0.7492,0.7646) | 0.9398 (0.9332,0.9463) | 0.8385 (0.8323,0.8446) | 0.7443 (0.7346,0.7539) | 0.8590 (0.8541,0.8639) |
| | **14** | **0.8774 (0.8738,0.8809)** | **0.7660 (0.7613,0.7707)** | **0.9484 (0.9442,0.9525)** | **0.8475 (0.8429,0.8520)** | **0.7601 (0.7509,0.7691)** | **0.8663 (0.8626,0.8699)** |
| **Non-sex-special model** | 3 | 0.7337 (0.7269,0.7404) | 0.8072 (0.8020,0.8124) | 0.7262 (0.7196,0.7328) | 0.7646 (0.7590,0.7701) | 0.4634 (0.4497,0.4771) | 0.7280 (0.7210,0.7349) |
| | 5 | 0.7512 (0.7453,0.7571) | 0.8115 (0.8034,0.8195) | 0.7463 (0.7417,0.7508) | 0.7775 (0.7718,0.7832) | 0.4988 (0.4867,0.5109) | 0.7466 (0.7407,0.7524) |
| | 7 | 0.7891 (0.7853,0.7930) | 0.8097 (0.8033,0.8161) | 0.7993 (0.7946,0.8041) | 0.8045 (0.8007,0.8082) | 0.5759 (0.5680,0.5837) | 0.7876 (0.7837,0.7915) |
| | 9 | 0.8029 (0.7993,0.8066) | 0.8161 (0.8108,0.8215) | 0.8161 (0.8100,0.8222) | 0.8161 (0.8130,0.8192) | 0.6040 (0.5964,0.6115) | 0.8019 (0.7980,0.8058) |
| | 11 | 0.8041 (0.8025,0.8058) | 0.8163 (0.8078,0.8248) | 0.8178 (0.8140,0.8217) | 0.8170 (0.8144,0.8197) | 0.6065 (0.6034,0.6095) | 0.8032 (0.8019,0.8045) |
| | **14** | **0.8095 (0.8064,0.8127)** | **0.8253 (0.8187,0.8319)** | **0.8217 (0.8177,0.8257)** | **0.8226 (0.8193,0.8259)** | **0.6179 (0.6101,0.6258)** | **0.8085 (0.8053,0.8117)** |

(4) A new biomarker ΔAREA is proposed to improve classification accuracy; (5) Sex-specific classifiers outperformed non-sex-specific classifiers, while sex-biased classifiers were inferior to sex-specific classifiers; and (6) The sex-dependent differences in drug response may be partly explained by lower levels of IK1, INa, and Ito in female versus male patients. These results underscore the importance of incorporating sex differences in the evaluation of AAD efficacy and emphasize the potential for personalized medicine approaches in treating AF based on AAD classification. This study indicates the value of combining mechanistic simulations with ML and provides insight into the role of gender factor in the drug efficacy evaluation.

## 3.1. Ionic mechanisms underlying sex differences in action potentials

To our knowledge, this is the first study to create a virtual AF population that considers sex disparities. We generated sex-specific models by parameterizing the Grandi model of human atrial electrophysiology, incorporating differences in

**Table 4. Principal Components (PCs) Analysis Coupled with Support Vector Machine (SVM) Classification Utilizing Sex-Specific Classifiers (Male and Female) for Discriminating Class III Drugs from Non-Class III Drugs.**

| Type | Number | ACC | SEN | SPE | F1 | AUC | MCC |
|---|---|---|---|---|---|---|---|
| Sex-special model (Male) (PCA) | 3 | 0.7594 (0.7561,0.7627) | 0.6834 (0.6729,0.6939) | 0.7329 (0.7278,0.7381) | 0.7072 (0.7018,0.7126) | 0.7495 (0.7457,0.7533) | 0.5044 (0.4974,0.5114) |
| | 5 | 0.8005 (0.7975,0.8035) | 0.6956 (0.6833,0.7079) | 0.8088 (0.8035,0.8141) | 0.7478 (0.7419,0.7537) | 0.7869 (0.7828,0.7909) | 0.5890 (0.5828,0.5953) |
| | 7 | 0.8120 (0.8080,0.8161) | 0.7151 (0.7004,0.7298) | 0.8201 (0.8135,0.8267) | 0.7639 (0.7568,0.7710) | 0.7994 (0.7943,0.8046) | 0.6131 (0.6048,0.6214) |
| | 9 | 0.8376 (0.8347,0.8405) | 0.7221 (0.7121,0.7321) | 0.8744 (0.8692,0.8797) | 0.7909 (0.7859,0.7959) | 0.8226 (0.8189,0.8263) | 0.6684 (0.6626,0.6742) |
| | 10 | 0.8725 (0.8691,0.8759) | 0.7470 (0.7371,0.7569) | 0.9412 (0.9377,0.9446) | 0.8328 (0.8274,0.8383) | 0.8562 (0.8520,0.8604) | 0.7448 (0.7385,0.7512) |
| | 11 | 0.8815 (0.8760,0.8871) | 0.76243 (0.7506,0.7741) | 0.9491 (0.9432,0.9550) | 0.8455 (0.8375,0.8535) | 0.8660 (0.8598,0.8723) | 0.7632 (0.7522,0.7743) |
| | **12** | **0.8816 (0.8761,0.8870)** | **0.7625 (0.7509,0.7741)** | **0.9491 (0.9432,0.9550)** | **0.8456 (0.8377,0.8535)** | **0.8661 (0.8599,0.8723)** | **0.7633 (0.7524,0.7743)** |
| | 13 | 0.8816 (0.8761,0.8870) | 0.7625 (0.7509,0.7741) | 0.9491 (0.9432,0.9550) | 0.8456 (0.8377,0.8535) | 0.8661 (0.8599,0.8723) | 0.7633 (0.7524,0.7743) |
| | 14 | 0.8816 (0.8761,0.8870) | 0.7625 (0.7509,0.7741) | 0.9491 (0.9432,0.9550) | 0.8456 (0.8377,0.8535) | 0.8661 (0.8599,0.8723) | 0.7633 (0.7524,0.7743) |
| Sex-special model (Female) (PCA) | 3 | 0.7520 (0.7417,0.7624) | 0.6739 (0.6578,0.6899) | 0.7467 (0.7349,0.7585) | 0.7084 (0.6950,0.7217) | 0.7445 (0.7337,0.7554) | 0.4958 (0.4744,0.5171) |
| | 5 | 0.7944 (0.7858,0.8030) | 0.7051 (0.6944,0.7158) | 0.8105 (0.7969,0.8241) | 0.7541 (0.7440,0.7641) | 0.7858 (0.7772,0.7944) | 0.5831 (0.5652,0.6010) |
| | 7 | 0.8048 (0.7971,0.8126) | 0.7137 (0.7061,0.7213) | 0.8263 (0.8107,0.8420) | 0.7658 (0.7577,0.7739) | 0.7961 (0.7887,0.8035) | 0.6049 (0.5884,0.6215) |
| | 9 | 0.8493 (0.8420,0.8566) | 0.7327 (0.7259,0.7395) | 0.9130 (0.9010,0.9250) | 0.8130 (0.8042,0.8217) | 0.8381 (0.8308,0.8453) | 0.7010 (0.6854,0.7166) |
| | 10 | 0.8529 (0.8437,0.8622) | 0.7352 (0.7251,0.7452) | 0.9198 (0.9060,0.9337) | 0.8172 (0.8058,0.8286) | 0.8417 (0.8323,0.8510) | 0.7090 (0.6894,0.7287) |
| | **11** | **0.8793 (0.8744,0.8842)** | **0.7680 (0.7604,0.7756)** | **0.9529 (0.9480,0.9577)** | **0.8505 (0.8441,0.8569)** | **0.8687 (0.8635,0.8738)** | **0.7636 (0.7537,0.7735)** |
| | 12 | 0.8793 (0.8744,0.8842) | 0.7680 (0.7604,0.7756) | 0.9529 (0.9480,0.9577) | 0.8505 (0.8441,0.8569) | 0.8687 (0.8635,0.8738) | 0.7636 (0.7537,0.7735) |
| | 13 | 0.8793 (0.8744,0.8842) | 0.7680 (0.7604,0.7756) | 0.9529 (0.9480,0.9577) | 0.8505 (0.8441,0.8569) | 0.8687 (0.8635,0.8738) | 0.7636 (0.7537,0.7735) |
| | 14 | 0.8793 (0.8744,0.8842) | 0.7680 (0.7604,0.7756) | 0.9529 (0.9480,0.9577) | 0.8505 (0.8441,0.8569) | 0.8687 (0.8635,0.8738) | 0.7636 (0.7537,0.7735) |

the expression levels of ion channels and transporters. The resulting parameterizations successfully recapitulated the well-documented differences in AP properties found in the literature [58]: RMP is less negative, APA and dV/dt_max are smaller, and APD is longer in female atrial cells compared to male atrial cells. These sex differences in APs are associated with the ion current differences between male and female AF subpopulations. Among the fourteen parameterized ion currents, we found that INa, Ito, IKs, IKur, IK1, and ICaL are smaller, and INaCa, Irel, and Ileak are larger in female versus male AF subpopulations, with statistical significance ($p < 0.05$, Table 3). Our findings are supported by previous experimental studies that have documented sex differences in ion currents across various animal models. Specifically: ① Smaller IK1, ICaL and delayed rectifier potassium current were observed in isolated ventricular myocytes from female versus male rabbit [59] and guinea pig hearts [60]; ② Smaller Ito was found in isolated ventricular myocytes from adult female

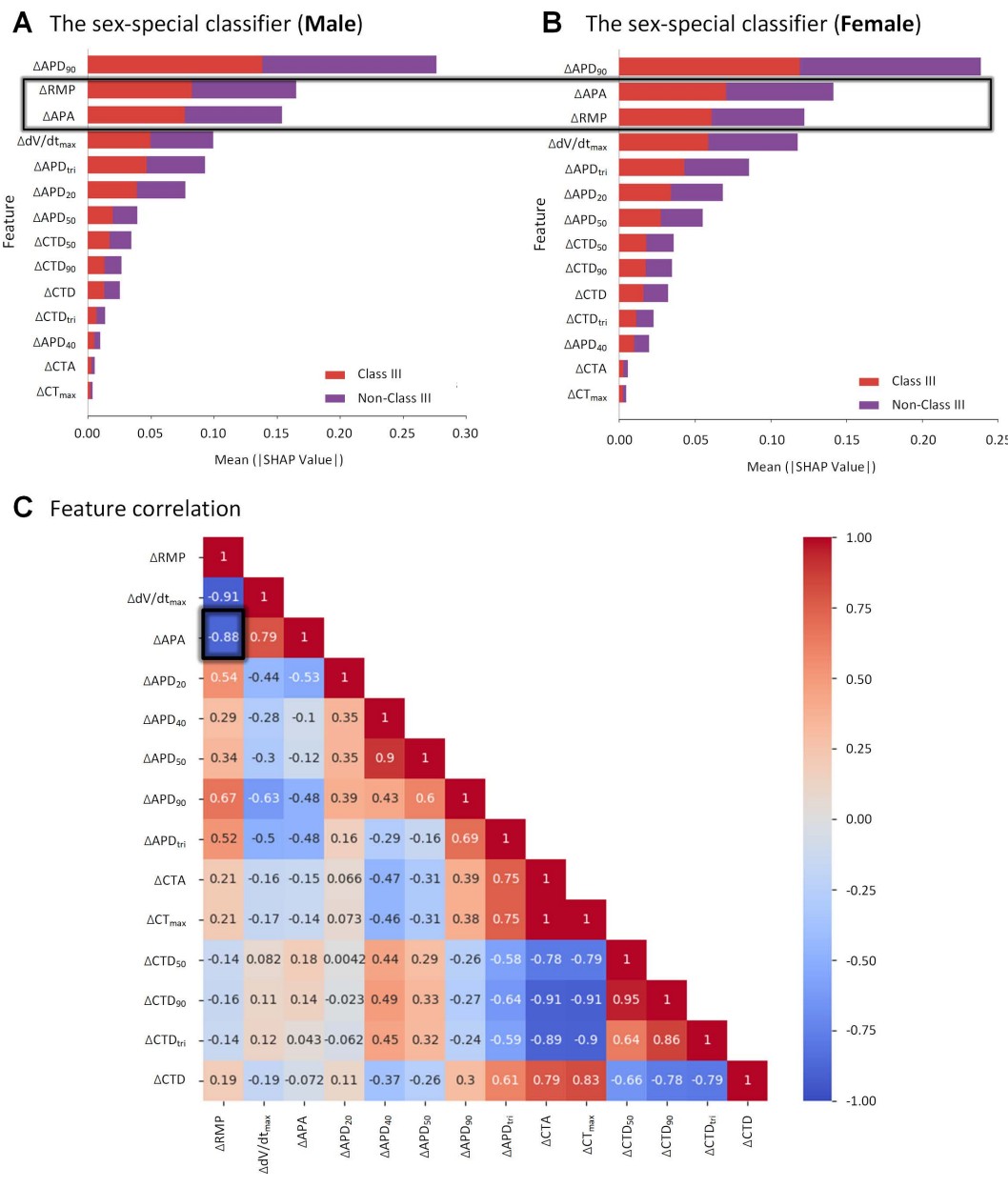

**Fig 9. Feature analysis of sex-specific classifiers.** Feature importance for the male (A) and female (B) classifiers is measured by the SHAP value. Feature correlation analysis shows that the two key features have a high degree of negative correlation (C).

mice compared to their male littermates [61]; ③ Smaller INa amplitude was detected in the female epicardial and endo-cardial layers of the left ventricle, but was similar to males in the mid-myocardium [62]; ④ A significantly lower expression of Kv1.5 and its corresponding IKur was observed in female versus male mouse ventricles [63]; and ⑤ Sex hormones, particularly estrogen, have been shown to affect calcium handling properties, contributing to atrial arrhythmogenesis by increasing sodium-calcium exchanger activity [64], RyR2 leakiness [65], and higher calcium spark frequency [66,67]. While these observations are predominantly from studies on animal ventricular cells, there is a scarcity of comparable data on human atrial cells. Furthermore, some results differ between studies. For instance, no statistical difference in ICaL

**Table 5. Feature ablation experimental results.**

| Type | features | ACC | SEN | SPE | F1 | AUC | MCC |
|------|----------|-----|-----|-----|-----|-----|-----|
| Sex-special model (Male) | W/oΔRMP | 0.8437 (0.8347,0.8526) | 0.7316 (0.7206,0.7427) | 0.8823 (0.8609,0.9036) | 0.7997 (0.7897,0.8097) | 0.8293 (0.8210,0.8376) | 0.6815 (0.6620,0.7010) |
| | W/oΔAPA | 0.8542 (0.8477,0.8606) | 0.7421 (0.7384,0.7458) | 0.8986 (0.8831,0.9142) | 0.8128 (0.8059,0.8198) | 0.8398 (0.8340,0.8456) | 0.7038 (0.6895,0.7182) |
| | – | 0.8802 (0.8751,0.8853) | 0.7635 (0.7569,0.7701) | 0.9454 (0.9320,0.9587) | 0.8450 (0.8389,0.8510) | 0.8659 (0.8607,0.8703) | 0.7620 (0.7478,0.7762) |
| Sex-special model (Female) | W/oΔRMP | 0.8374 (0.8313,0.8434) | 0.7485 (0.7379,0.7591) | 0.8677 (0.8582,0.8772) | 0.8037 (0.7960,0.8113) | 0.8285 (0.8223,0.8348) | 0.6715 (0.6591,0.6840) |
| | W/oΔAPA | 0.8234 (0.8193,0.8274) | 0.7210 (0.7159,0.7262) | 0.8594 (0.8490,0.8697) | 0.7841 (0.7801,0.7881) | 0.8132 (0.8095,0.8169) | 0.6437 (0.6348,0.6526) |
| | – | 0.8774 (0.8738,0.8809) | 0.7660 (0.7613,0.7707) | 0.9484 (0.9442,0.9525) | 0.8475 (0.8429,0.8520) | 0.8663 (0.8626,0.8699) | 0.7601 (0.7509,0.7691) |

**Table 6. Role of ΔRMP or ΔAPA in the sex-special models.**

| Type | features | ΔACC (%) | ΔSEN (%) | ΔSPE (%) | ΔF1 (%) | ΔAUC (%) | ΔMCC (%) |
|------|----------|----------|----------|----------|---------|----------|----------|
| Sex-special model (Male) | **W/oΔRMP** | **3.7%** | **3.2%** | **6.3%** | **4.5%** | **3.7%** | **8.1%** |
| | W/oΔAPA | 2.6% | 2.1% | 4.7% | 3.2% | 2.6% | 5.8% |
| Sex-special model (Female) | W/oΔRMP | 4.0% | 1.8% | 8.1% | 4.4% | 3.8% | 8.9% |
| | **W/oΔAPA** | **5.4%** | **4.5%** | **8.9%** | **6.3%** | **5.3%** | **11.6%** |

density was found in human atrial cells [67], whereas ICaL was reported to be 32% higher in isolated ventricular myocytes from female rabbits compared to males [59]. These discrepancies underscore the need for further biological experiments to verify differences in human atrial ion currents.

## 3.2. Sex-specific classifiers of antiarrhythmic drugs for atrial fibrillation

To investigate the impact of sex differences on drug responses and classification in AF patients, we proposed the adoption of sex-specific AAD classifiers. Using the GridSearchCV algorithm to optimize hyperparameters for maximum classification performance [68], we developed male and female classifiers based on features representing changes in AP and CT biomarkers before and after medication. Our results demonstrated that SVM performed best among the six classic ML models, improving classification performance by incorporating the gender factor. Analysis of predictive biomarkers revealed that AP biomarkers are more important than CT biomarkers for recognizing Class III AADs. Notably, two AP features—ΔRMP and ΔAPA—contribute differently in sex-specific classifiers for males and females. A high degree of correlation between ΔRMP and ΔAPA was observed, with a correlation coefficient of -0.88. This finding supports previous studies indicating a trend toward a less negative RMP and smaller APA in human atrial cells from women [58]. Our parameter sensitivity analysis further supported this: RMP is highly correlated with the inward rectifier potassium current IK1, while APA is also correlated with IK1, INa, and Ito. Our results suggest that lower APA can be partially explained by less negative RMP, indicating that less INa [62], Ito [69] and IK1 [70,71] in women may be the relevant sex-dependent differences. Randomized studies have shown that female gender is an independent predictor of AF recurrence after catheter ablation (AFCA), and that AAD response is better in elderly women than in men with AFCA [72], and that AAD response is better in elderly women than in men with AFCA [73]. These sex differences may impact the development and screening of new drugs and personalized medicine.

**Table 7. Effects of the new geometric biomarker on classification performance.**

| Type | Models | ACC | SEN | SPE | F1 | MCC | AUC |
|---|---|---|---|---|---|---|---|
| **Sex-special model (Male)** | LR | 0.7471 (0.7412,0.7529) | 0.6849 (0.6683,0.7014) | 0.7122 (0.7084,0.7159) | 0.6982 (0.6881,0.7082) | 0.4810 (0.4680,0.4940) | 0.73922 (0.7320,0.7463) |
| | SVM | **0.8922 (0.8878,0.8967)** | 0.8551 (0.8405,0.8698) | **0.8886 (0.8848,0.8924)** | **0.8715 (0.8650,0.8779)** | **0.7794 (0.7701,0.7886)** | **0.8875 (0.8819,0.8932)** |
| | NB | 0.6316 (0.6249,0.6384) | **0.9404 (0.9324,0.9484)** | 0.5396 (0.5350,0.5442) | 0.6857 (0.6801,0.6913) | 0.3877 (0.3725,0.4028) | 0.6708 (0.6640,0.6776) |
| | XGB | 0.8847 (0.8788,0.8907) | 0.8499 (0.8374,0.8623) | 0.8767 (0.8710,0.8824) | 0.8630 (0.8554,0.87075) | 0.7640 (0.7517,0.7762) | 0.8803 (0.8737,0.8869) |
| | RF | 0.8660 (0.8583,0.8737) | 0.8317 (0.8190,0.8443) | 0.8515 (0.8399,0.8632) | 0.8414 (0.8322,0.8506) | 0.7257 (0.7100,0.7415) | 0.8616 (0.8537,0.8695) |
| | KNN | 0.8504 (0.8460,0.8547) | 0.8179 (0.8049,0.8309) | 0.8297 (0.8232,0.8362) | 0.8237 (0.8175,0.8298) | 0.6940 (0.6847,0.7032) | 0.8463 (0.8411,0.8514) |
| **Sex-special model (Female)** | LR | 0.7262 (0.7195,0.7330) | 0.6833 (0.6749,0.6917) | 0.6995 (0.6903,0.7087) | 0.6913 (0.6843,0.6983) | 0.4456 (0.4323,0.4590) | 0.7222 (0.7157,0.7288) |
| | SVM | **0.8905 (0.8866,0.8943)** | 0.8535 (0.8462,0.8609) | **0.8972 (0.8942,0.9003)** | **0.8748 (0.8701,0.8796)** | **0.7784 (0.7705,0.7862)** | **0.8870 (0.8828,0.8912)** |
| | NB | 0.6207 (0.6125,0.6288) | **0.9168 (0.9115,0.9221)** | 0.5459 (0.5404,0.5515) | 0.6844 (0.6786,0.6902) | 0.3422 (0.3257,0.3587) | 0.6483 (0.6405,0.6562) |
| | XGB | 0.8772 (0.8745,0.8798) | 0.8557 (0.8520,0.8594) | 0.8685 (0.8644,0.8726) | 0.8621 (0.8591,0.8650) | 0.7515 (0.7461,0.7569) | 0.8752 (0.8725,0.8778) |
| | RF | 0.8598 (0.8541,0.8654) | 0.8326 (0.8243,0.8408) | 0.8515 (0.8443,0.8587) | 0.8419 (0.8354,0.8484) | 0.7162 (0.7046,0.7277) | 0.8572 (0.8514,0.8630) |
| | KNN | 0.8447 (0.8387,0.8507) | 0.8130 (0.7997,0.8263) | 0.8363 (0.8301,0.8425) | 0.8244 (0.8167,0.8322) | 0.6857 (0.6733,0.6980) | 0.8418 (0.8352,0.8483) |
| **Non-sex-special model** | LR | 0.6774 (0.6731,0.6817) | 0.8479 (0.8415,0.8544) | 0.6562 (0.6528,0.6597) | 0.7397 (0.7360,0.7433) | 0.3441 (0.3347,0.3534) | 0.6642 (0.6598,0.6686) |
| | SVM | **0.8316 (0.8292,0.8341)** | 0.8554 (0.8470,0.8638) | **0.8347 (0.8275,0.8419)** | **0.8448 (0.8428,0.8469)** | **0.6613 (0.6563,0.6663)** | **0.8298 (0.8270,0.8327)** |
| | NB | 0.6565 (0.6519,0.6611) | **0.9041 (0.8975,0.9108)** | 0.6238 (0.6210,0.6266) | 0.7382 (0.7344,0.7421) | 0.3290 (0.3174,0.3406) | 0.6373 (0.6328,0.6419) |
| | XGB | 0.8287 (0.8253,0.8321) | 0.8476 (0.8418,0.8533) | 0.8349 (0.8306,0.8392) | 0.8412 (0.8379,0.8444) | 0.6554 (0.6486,0.6622) | 0.8272 (0.8238,0.8306) |
| | RF | 0.8050 (0.8000,0.8101) | 0.8237 (0.8116,0.8357) | 0.8146 (0.8110,0.8182) | 0.8190 (0.8132,0.8249) | 0.6079 (0.5979,0.6179) | 0.8036 (0.7989,0.8083) |
| | KNN | 0.7992 (0.7949,0.8036) | 0.8539 (0.8464,0.8614) | 0.7890 (0.7814,0.7965) | 0.8201 (0.8168,0.8234) | 0.5964 (0.5878,0.6049) | 0.7950 (0.7902,0.7998) |

### 3.3. Comparison with previous modelling studies

Unlike previous modeling studies that relied on idealized systems or animal-derived parameters [74,75], our human-centric framework is uniquely grounded in patient-specific electrophysiological data that intrinsically captures integrated clinical variables (age, comorbidities, polypharmacy) [58]. This human-data-driven approach reveals fundamental sex-specific electrophysiological phenotypes: female cardiomyocytes exhibit a less negative RMP and demonstrate attenuated age-related APD90. These findings enable us to model sex as a biological variable with mechanistic underpinnings rather than merely as a statistical covariate, providing novel insights into sex-divergent pharmacodynamics and paving the way for precision medicine strategies.

Our integrative methodology represents a paradigm shift from traditional reductionist approaches through the synergistic combination of: (1) high-fidelity in vitro patch-clamp measurements, (2) population-based in silico modeling, and

## A  Parameter sensitivity

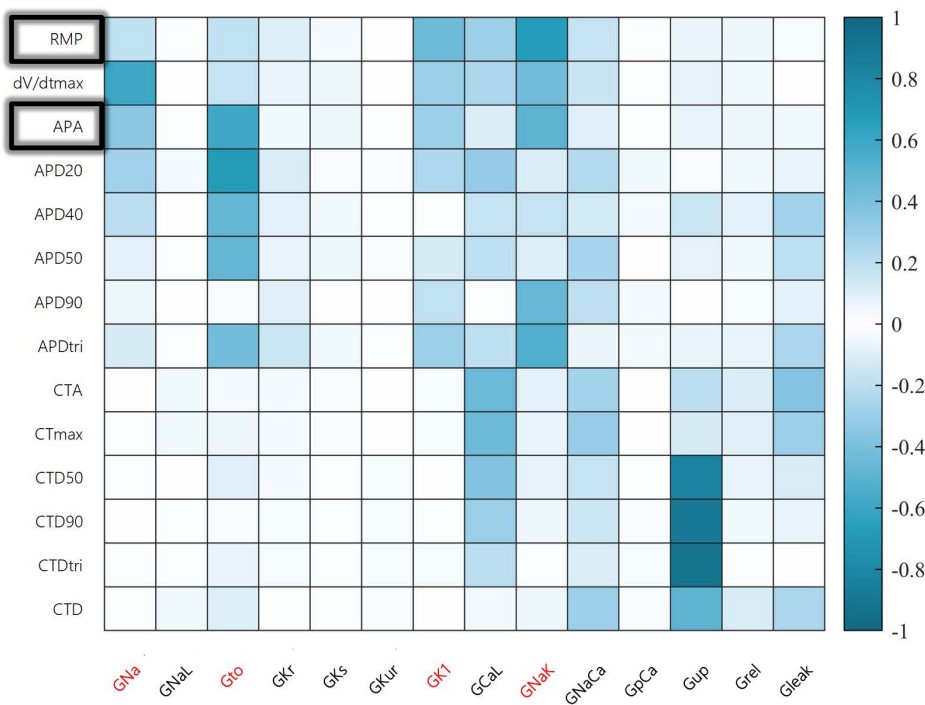

## B  Sex differences in ionic current parameters

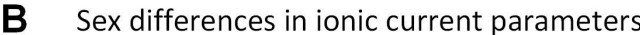
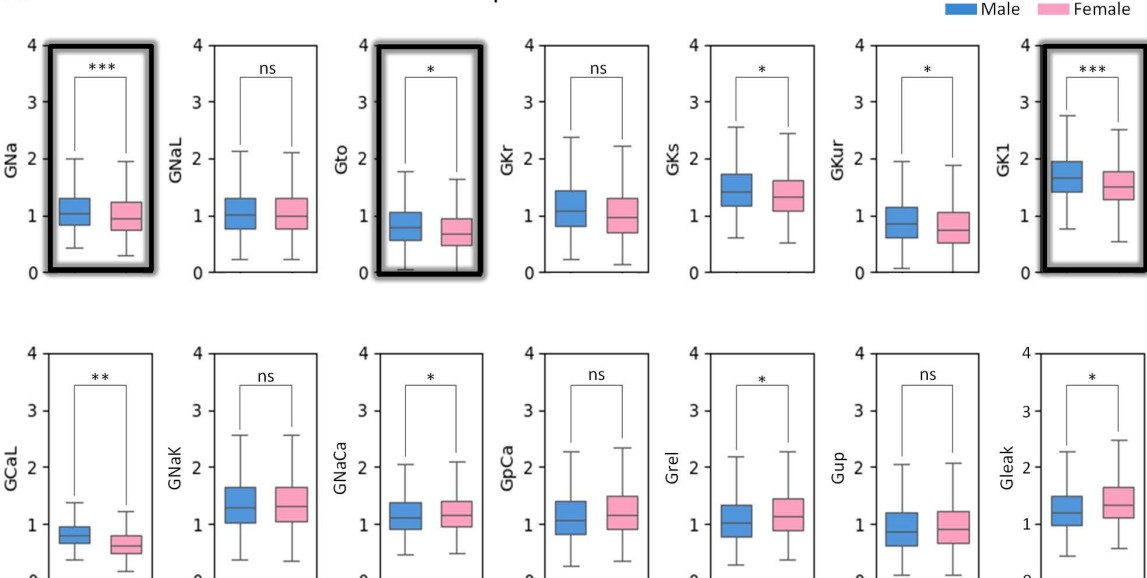

**Fig 10. Parameter analysis of sex differences.** (A) Sensitivity analysis via ion current parameters (x-axis) versus the model biomarkers (y-axis). (B) Boxplots of ion current parameters in males versus females. The p-values are indicated and the stars *, **, *** indicate a p-value < 0.05, 0.01 and 0.001 respectively. ns indicates not significant.

(3) explainable machine learning frameworks. While previous investigations have employed isolated components of this pipeline [76–78], our unified approach provides three key advantages: First, the in silico platform enables probabilistic

simulation of drug effects across a physiologically diverse cardiomyocyte population, capturing inter-subject variability that single-cell experiments cannot address. Second, our SHAP-based feature importance analysis identifies non-linear interaction effects between clinical variables and drug responses. Third, the in vitro data serve not merely for validation, but as the foundational training set for our machine learning models, creating a closed-loop system that iteratively improves predictive accuracy.

The derived ΔAREA biomarker represents a conceptual advance over conventional parameters by quantifying action potential topology through a novel geometric integration of ΔRMP, ΔAPA, and ΔAPD90. This multidimensional metric: (1) improves classification AUC for drug-induced proarrhythmia by ~2.1%, (2) ranks as the fourth most predictive feature in our XAI framework, and (3) maintains clinical interpretability through direct correspondence with established electrophysiological principles. Unlike prior biomarkers focusing on isolated parameters, ΔAREA captures the dynamic interaction between depolarization and repolarization abnormalities - a critical advantage given the established limitations of single-parameter biomarkers in clinical risk stratification.

### 3.4. The potential applications in clinical practice

This study has practical applications in clinical practice, drug development, and understanding disease mechanisms [79–81]. By providing a more comprehensive understanding of gender differences in drug response, it can lead to more personalized and effective medical care [82].

In clinical practice, the ability to accurately predict a drug's efficacy and potential adverse reactions is crucial for prescribing the right medication to each patient [83,84]. The sex-specific classifiers developed in this study can provide valuable insights that go beyond simply looking at the drug's mode of action. By considering gender differences, clinicians can make more informed decisions when prescribing antiarrhythmic drugs. For example, knowing that female patients are more likely to experience certain adverse drug reactions due to differences in ion currents and calcium handling properties [85], doctors can adjust the dosage or choose a different drug altogether. This approach can lead to better patient outcomes, reduced risk of adverse events, and more effective medical care.

The pipeline developed in this study can enhance the drug development and screening process. By accurately classifying drugs based on their effects on different patient populations, researchers can identify potential candidates for further development more efficiently [86]. Additionally, understanding the gender-specific differences in drug response can help in tailoring drug formulations and clinical trials to better meet the needs of different patient groups. This can lead to the development of more targeted and effective drugs, reducing the time and cost of drug development.

Studying the ionic mechanisms underlying gender differences in drug response can contribute to a better understanding of the pathophysiology of atrial fibrillation and other cardiac rhythm disorders [4]. This knowledge can lead to the development of new therapeutic strategies and interventions that take into account gender-specific factors. For example, understanding the role of estrogen in calcium handling properties can help in developing drugs that target these specific pathways, potentially reducing the risk of atrial arrhythmogenesis [87].

### 3.5. Limitations

There are several limitations specific to this study.

Firstly, the current study's binary classification task, which divided drugs into Class III and non-Class III (while these drugs can be further categorized into more than four according to the Vaughan Williams classification system), was rather limiting. The focus on Class III drugs was due to their extensive use in AF and potential risk of inducing TdP, a dangerous form of polymorphic ventricular tachycardia. Our intention was to perform a fundamental proof of concept using as the first step the most straightforward (binary) classification and formulate a methodological framework that could serve as a springboard for subsequent investigations. From a clinical application perspective, further research is required to explore more detailed classification methods. Future work should expand the drug classification scope to include a more diverse

PLOS Computational Biology

range of drug categories and develop more refined criteria. This should consider not only the drug's mechanism of action but also its efficacy and safety in different patient populations, especially in relation to gender differences.

Secondly, we acknowledge that risk identification of drugs is more important than their classification. Previous studies have conducted risk assessments of AADs for AF based on computer models. However, unlike AADs for ventricular tachycardia, these AADs for AF do not have risk levels suggested by the Comprehensive in vitro Proarrhythmia Assay (CiPA) categorization system, making it possible to develop and validate a mechanistic-based assessment of the proarrhythmic risk of drugs for ventricular tachycardia. Therefore, this study focuses on the role of gender factors in drug screening and personalized medication rather than on drug risk assessment. In future research, we aim to integrate drug efficacy and safety research more comprehensively. We will explore methods to combine the assessment of drug efficacy with the identification of drug risks, especially in the context of gender differences. This could involve developing more accurate risk prediction models that incorporate gender-specific biomarkers and physiological characteristics to better evaluate the overall benefit-risk ratio of drugs and provide more reliable guidance for clinical decision-making.

Thirdly, while AF pathophysiology critically involves calcium dysregulation and action potential shortening, the experimental data used to calibrate our computational model—though capturing integrated effects of clinical factors (age, BMI, hypertension, etc.) on cardiomyocyte electrophysiology—lacked explicit calcium transient measurements [58]. This precluded implementation of biochemically detailed calcium-handling systems and obscured potential gender differences in calcium dynamics (e.g., sex hormone modulation of SERCA/RyR2 activity). While population-level calibration implicitly accounts for comorbidities, we could not resolve how calcium dysregulation interacts with clinical variables (e.g., beta-blocker use) across sexes. Future studies will employ sex-stratified iPSC-CM calcium imaging and mechanistic modeling (e.g., CaMKII/RyR2 phosphorylation states) to bridge this gap, enabling more precise evaluation of drug-gender interactions in AF.

Finally, the electrophysiological phenomena of single atrial cells can only partially reflect the reentrant arrhythmias at the organ level in AF. At the organ level, AF may involve gender differences in cardiac anatomical structure [88–90], the degree and distribution of fibrosis [91], and potential organ-level electrical heterogeneity. Therefore, comprehensive gender differences in the evaluation of drug efficacy and safety require more in-depth research. In future investigations, we plan to adopt a more integrated approach that combines single-cell studies with organ-level models. This could involve using in vivo animal models or computational simulations of the whole heart to better understand the complex interactions between drugs, gender differences, and the organ-level electrophysiological environment [92]. By doing so, we hope to obtain a more comprehensive understanding of the relationship between drug efficacy and safety and gender differences at the organ level and translate these findings into more effective clinical strategies.

Nevertheless, while it is important to explicitly state the potential limitations of the approaches adopted in this study, these potential limitations are not expected to influence the fundamental conclusions that can be drawn from our data. Instead, they serve as a foundation for future research directions and improvements, guiding us to further enhance the quality and impact of our work in the field of drug research and personalized medicine.

## 4. Conclusion

We developed a pipeline which for the first time allows to identify true gender-specific Class III aniarrhythmic drugs based on in vitro measurements, in silico models and machine learning tools. It includes a population of calibrated sex-specific models which can be used to simulate the effects of drugs on action potential (AP) and calcium transient (CT). From these simulations, biomarkers are extracted and based on those trained ML classifiers can predict true Class III aniarrhythmic drugs. Sex-specific classifiers significantly outperformed non-sex-specific classifiers in predicting drug efficacy, highlighting the importance of considering gender in drug evaluation. We also studied which biomarkers are the most significant for differentiating male and female responses to AADs, and also found that lower levels of IK1, INa, and Ito in female patients

may contribute to these differences. Overall, our study reveals the potential of computational models and ML in enhancing drug screening processes and developing personalized treatment strategies.

## 5. Materials and methods

### 5.1. Ethics statement

The details of obtaining the experimental dataset are provided in [58]. We reiterate them here for completeness.

The experimental datasets were obtained in studies with human samples conforming to the Declaration of Helsinki. All investigations were approved by the Medical Faculty Ethics Committee of the Technical University Dresden; approval number EK 114082202. Each patient gave written, informed consent.

**Dataset.** Clinical parameters were collected from the medical records of patients and analyzed after anonymization [58]. Data were retrospectively analyzed from 201 patients with longstanding persistent AF were included in this analysis. After excision, tissue was immediately placed at room temperature into a non-oxygenated cardioplegic solution (in mM): NaCl 100, taurine 50, glucose 20, KCl 10, $MgSO_4$ 5, MOPS (3-(N-morpholino) propanesulfonic acid) 5, $KH_2PO_4$ 1.2, containing 30 mM of the myosin ATPase inhibitor BDM (2,3-butanedione monoxime) and transferred to the laboratory in < 10 min. Detailed information on clinical characteristics and medication is provided in the Table A in S2 Text.

APs were recorded with standard intracellular microelectrodes in atrial trabeculae (196 recordings from 201 patients: 180 recordings from 126 male AF patients and 107 recordings from 75 female AF patients). Bath solution contained (in mM): NaCl 127, KCl 4.5, MgCl2 1.5, CaCl2 1.8, glucose 10, NaHCO3 22, NaH2PO4 0.42, equilibrated with O2-CO2 [95:5] at $36.5 \pm 0.58°C$, pH 7.4. Preparations were regularly stimulated at 1 Hz for at least 1 h before data acquisition with a custom-made computer program (University of Szeged, Hungary) that also generated electrical stimuli. The following parameters were quantified to characterize inter-subject variability in human atrial AP: APD20, APD50, APD90, APA, RMP, AP plateau potential at 20% of APD20 (V20), and dV/dtmax [58]. The ranges of these biomarkers are presented in the Table A in S3 Text.

### 5.2. Human atrial electrophysiology cell model

The Grandi et al. model was used as a base to construct the computational AP model populations [93]. This model provides a biophysically-detailed description of human atrial cellular electrophysiology including main transmembrane ionic currents, including the fast sodium current (INa), the late sodium current (INaL), the L-type calcium current (ICaL), the transient outward potassium current (Ito), the ultra-rapid potassium current (IKur), the inward rectifier potassium current (IK1), the rapid and slow components of the delayed rectifier potassium current (IKr and IKs) and those associated with the sodium/potassium pump (INaK), the calcium pump (IpCa), the sodium/calcium exchanger (INaCa). This model also includes representations of intracellular calcium handling with sarcoplasmic reticulum calcium release (Irel), leak (Ileak), and uptake (Iup) currents, as well as ionic homeostasis regulating sodium, potassium, and calcium intracellular concentrations, which determine the time course of the human atrial AP. The Grandi et al. model has been reviewed in detail in previous publications [93], and here we provide a brief description on its main characteristics.

### 5.3. Atrial fibrillation population

To capture inter-subject variability, the population of sampled models of human atrial electrophysiology for AF was generated based on the Grandi model [93]. All models in the AF population shared the same equations, but the 14 ionic current conductances determining the human atrial AP were varied with respect to their original values. These parameters were the conductances of INa, INaL, ICaL, Ito, IKur, IKr, IKs, IK1, INaK, INaCa and IpCa, as well as the permeabilities of Irel, Ileak, and Iup.

Our first step is to estimate the median values and range of variation for the 14 ionic current conductances/permeabilities required to obtain simulated APs to be within the experimental range for each model [21]. To do so and to minimize computational expense, we first constructed coarse model populations with 100,000 different ionic conductance combinations

sampled over a variation range from -100% to +200% around their values in the original models, using Latin hypercube sampling. The Grandi default models were initially preconditioned by pacing at 1 Hz (using a 5 ms stimulus duration, −12.5 pA/pF amplitude) until the steady-state was reached (changes in state variables between consecutive stimuli measured at the end of each cardiac cycle smaller than 1%). All AP models within the populations were paced at 1 Hz, and APs were analyzed following a train of 500 periodic stimuli to quantify $APD_{20}$, $APD_{50}$, $APD_{90}$, APA, RMP, $V_{20}$ and $dV/dt_{max}$ for each model in each population (Fig A in S4 Text). As illustrated in the Fig A in S5 Text, we initiated with an initial population of 100,000 models. In the first step, 20,963 models with failed repolarization and 5,447 models with abnormal repolarization were excluded, leading to 73,590 models with normal repolarization. Subsequently, the population was further refined by eliminating 61,743 models that did not lie within the range of experimental measurements. The parameters for these experimental measurements and their specific ranges are detailed in the Table A in S3 Text. Among the excluded models, 29,700 did not match the characteristics of male atrial fibrillation, and 32,043 did not match those of female atrial fibrillation. Through this meticulous and multi-step screening procedure, we finally obtained 11,847 models, consisting of 5,663 male models and 6,184 female models. The distribution of each ion current in males and females is shown in the Fig A in S6 Text.

## 5.4. Antiarrhythmic drug population

To investigate different effects of AADs, the population of sampled models of human atrial electrophysiology for Class III and non-Class III AADs was generated based on models in the AF population. Class III AADs include Amiodarone, Dofetilide, Dronedarone, Ibutilide, Sotalol, and Vernakalant, while non-Class III AADs include Flecainide, Propafenone, Quinidine, Ranolazine, Disopyramide, and Digoxin [14]. A simple pore channel equation was used for drug implementation for each drug that was modeled according to experimental IC50 values to calculate the block of the channel (Table A in S7 Text) [52,94]. The model used is described as follows:

$$g_{i,drug} = g_i \left[ 1 + \left( \frac{D}{IC_{50,i}} \right)^h \right]^{-1}$$

(1)

where $g_{i,drug}$ is the maximal conductance of channel $i$ in the presence of the drug, $g_i$ is the initial conductance of the channel $i$ for each of the profiles in the AF population of modes, $D$ is the concentration of the drug and $IC_{50,i}$ is the concentration of the drug that reduces by 50% the channel current.

All AP models considering AAD effects in the AF population were paced at 1 Hz, and APs were analyzed following a train of 100 periodic stimuli to quantify AP and CT for each model in the Class III and non-Class III AAD subpopulations. Take a drug $i$ as an example. The initial population for this drug is also 11,847 models. However, among these models, $A_i$ models with abnormal repolarization are excluded. The number of models included in the drug population for drug is thus 11,847-$A_i$. In our study, we investigated a total of 12 drugs ($i = 1, 2, \ldots 12$). To obtain the final drug population, we summed up the populations for each drug. The final drug population is calculated as (11,847x12-($A_1 + A_2 + \ldots + A_{12}$)). Finally, 60,599 AP models were generated to form the ADD population, containing 31,842 AP models (15,221 male and 16,621 female AP models) in the Class III AAD subpopulation and 28,757 AP models (13,746 male and 15,011 female AP models) in the non-Class III subpopulation (Fig A in S8 Text and Table A in S9 Text). The proportion of AP models for each AAD ranged from 7% to 24%. Values of biomarkers of the AAD population can be found in the Table A in S10 Text for Class III vs. non-Class III and in the Table A in S11 Text for virtual male vs. female, respectively.

## 5.5. Feature extraction

We extracted 14 biomarkers for each sample in the generated AF and AAD populations. In these biomarkers, 8 biomarkers extracted from the AP recording included RMP, $dV/dt_{max}$, APA, $APD_{20}$, $APD_{40}$, $APD_{50}$, $APD_{90}$, AREA, and $APD_{tri}$ (Fig A in S12 Text), while 6 biomarkers extracted from the CT recording included $CT_{max}$, $CTD_{50}$, $CTD_{90}$, $CTD_{tri}$ and CTD (Fig A in

). The differences between the same biomarkers before (i.e., AF) and after medication (i.e., Class III or non-Class III AADs) were calculated as features and normalized to a Gaussian distribution using StandardScaler due to different scales [95].

## 5.6. Machine learning models

A total of 60,599 recordings (each containing 14 extracted features) corresponding to inter-subject variability in APs, gender differences (male and female conditions) and two conditions (Class III and non-Class III conditions) creates a database with different ionic conductance combinations. Six classic ML models, including Logistic Regression (LR), Support Vector Machine (SVM), Naïve Bayes (NB), Extreme Gradient Boosting (XGBoost), Random Forest (RF), and K-Nearest Neighbor (KNN), were trained with 70% of the data (n = 42,419) and 30% of the data was used for testing with a five-fold cross-validation. Each ML model was optimized to be suitable for this drug classification task by using the GridSearchCV algorithm to find the combination of hyperparameters that maximizes its AUC on the test data set [96]. These hyperparameters and corresponding values of each ML model are detailed in the Table A in S13 Text.

## 5.7. Evaluation metrics

The performance of ML models was evaluated from two aspects. ① Non-quantitative assessment: Feature importance analysis evaluated the contribution of each feature to the classification performance of the ML model by calculating SHAP values; and ② quantitative assessment was performed by using the following metrics: accuracy (ACC), sensitivity (SEN), specificity (SPE), F1 score (F1), area under the receiver operating characteristic (ROC) curve (AUC) and Mathew Correlation Coefficient (MCC) [97]. They are defined as

$$ACC = \frac{TP + TN}{TP + FN + TN + FP} \tag{2}$$

$$SEN = \frac{TP}{TP + FN} \tag{3}$$

$$SPE = \frac{TN}{TN + FP} \tag{4}$$

$$F1 = \frac{2 * TP}{(2 * TP + FP + FN)} \tag{5}$$

$$MCC = (TN * TP - FP * FN)/\sqrt{(TP + FP)(TP + FN)(TN + FP)(TN + FN)} \tag{6}$$

where true positive (TP) represents the Class III sample being classified as the Class III sample and true negative (TN) represents non-Class III sample being classified as non-Class III sample. The false positive (FP) represents non-Class III samples being miss-classified as Class III sample and false negative (FN) represents Class III sample being miss-classified as Class III sample. The area under the curve (AUC) score was calculated by measuring the area under receiver operating characteristic (ROC) curve to assesses the model's ability in distinguish between different classes.

## 5.8. Statistical analysis

Characteristics were compared between groups using Student's t-test or Fisher's exact test, as appropriate. Hazard ratios and corresponding 95% confidence intervals were calculated using conditional logistic regression models. All statistical tests were performed using a significance level of $p < 0.05$. GraphPad Prism 10 (GraphPad Inc., Boston, MA, USA) was used for all analyses. All tests utilized were appropriate for the data.

## 5.9. Simulation tools

Modeling and simulations were conducted on desktop servers. The simulations used to create the population in the project were implemented using the stiff ordinary differential equation solver ode15s in MATLAB 2022b. The ML models were implemented in Python, using the packages NumPy, Pandas, Scikit-Learn, SHAP, and Pickle. The source code used in this study can be downloaded from the GitHub website (https://github.com/xuanyuanzaishui/Sex-Specific-Classification-of-Antiarrhythmic-Drugs).

## Supporting information

**S1 Text. Feature importance analysis.**
(DOCX)

**S2 Text. Patient characteristics.**
(DOCX)

**S3 Text. Values of biomarkers of action potential recordings from AF patients (male vs. female).**
(DOCX)

**S4 Text. The flowchart illustrates the process of constructing AF populations of males and females.**
(DOCX)

**S5 Text. The flowchart illustrates the model screening process of constructing AF populations of males and females. AP, action potential.**
(DOCX)

**S6 Text. Histograms of the scaling factors for the different ionic currents of the selected individuals (5,663 males and 6,184 females).**
(DOCX)

**S7 Text. The list of antiarrhythmic drugs with their concentration(μM), IC50 (μM) and hill coefficient.**
(DOCX)

**S8 Text. The distribution of drug populations.**
(DOCX)

**S9 Text. The model screening process of constructing drug populations of males and females.**
(DOCX)

**S10 Text. Values of biomarkers of the AAD population (Class III vs. non-Class III).**
(DOCX)

**S11 Text. Values of biomarkers of the AAD population (virtual male vs. female).**
(DOCX)

**S12 Text. The illustration of biomarkers extracted for each sample in the populations.**
(DOCX)

**S13 Text. Description of machine learning algorithms.**
(DOCX)

## Author contributions

**Conceptualization:** Weishan Wang, Hua Lu.

**Data curation:** Jieyun Bai, Weishan Wang.

**Formal analysis:** Jieyun Bai, Weishan Wang, Henggui Zhang, Alexander V Panfilov, Jichao Zhao.

**Funding acquisition:** Jieyun Bai, Alexander V Panfilov.

**Investigation:** Jieyun Bai, Weishan Wang, Xiaoshen Zhang, Hua Lu, Alexander V Panfilov, Jichao Zhao.

**Methodology:** Weishan Wang, Henggui Zhang.

**Project administration:** Jieyun Bai, Xiaoshen Zhang, Alexander V Panfilov, Jichao Zhao.

**Resources:** Jieyun Bai, Xiaoshen Zhang, Jichao Zhao.

**Software:** Weishan Wang.

**Supervision:** Hua Lu, Alexander V Panfilov, Jichao Zhao.

**Validation:** Jieyun Bai, Hua Lu, Henggui Zhang, Alexander V Panfilov, Jichao Zhao.

**Visualization:** Jieyun Bai, Weishan Wang.

**Writing – original draft:** Jieyun Bai, Weishan Wang, Xiaoshen Zhang, Alexander V Panfilov, Jichao Zhao.

**Writing – review & editing:** Jieyun Bai, Weishan Wang, Xiaoshen Zhang, Hua Lu, Henggui Zhang, Alexander V Panfilov, Jichao Zhao.

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
