## [Decision Letter · Decision Letter 0]

Dear Prof. Bai,

Thank you very much for submitting your manuscript "A Novel Pipeline for Identifying Gender-Specific Class III Antiarrhythmic Drugs Based on In Vitro Measurements, In-Silico Models and Machine Learning Tools" for consideration at PLOS Computational Biology.

As with all papers reviewed by the journal, your manuscript was reviewed by members of the editorial board and by several independent reviewers. In light of the reviews (below this email), we would like to invite the resubmission of a significantly-revised version that takes into account the reviewers' comments.

We cannot make any decision about publication until we have seen the revised manuscript and your response to the reviewers' comments. Your revised manuscript is also likely to be sent to reviewers for further evaluation.

Sincerely,

Yang Lu, Ph.D.

Academic Editor

PLOS Computational Biology

Jason Haugh

Section Editor

PLOS Computational Biology

Reviewer's Responses to Questions

**Comments to the Authors:**

Reviewer #1: see attachment

Reviewer #2: General Comment

This paper introduces sex-dependent machine learning (ML) algorithms for classifying whether the drug class that a patient with Atrial Fibrillation (AF) has taken belongs to Class III or not, based on action potential (AP) and calcium transient (CT) biomarkers. The results demonstrate that creating two classifiers for different sexes works better compared to a sex-non-specific classifier, and the AP features are more important than the CT features. The authors further tie in the relationship between different ion mechanisms to explain the gender difference in drug response. Overall, the paper is well written with extensive analysis, but the reviewer would still like to point out some issues to be addressed before proceeding with this work to publication.

Major Points

First, it seems concerning in section 2.1 that only 11,847 models were accepted from an initial population of 100,000 models. If the simulation only has around a 10% success rate in generating data that aligns with the experimentally determined ranges of biomarkers, it probably will require some further proof to demonstrate that the data generated from the simulation are indeed trustworthy to make the ML results more robust.

Similarly, the subsequent generation of drug populations needs some more explanation as well. The only line in the main text about this part of data generation is "By incorporating drug effects on different ion channels, we created an initial population from the AF population and removed models with abnormalities to generate a drug population of 60,599 AP models." It is not clear to the reviewer how the previously sampled 11,847 models can be augmented to 60,599 samples and what abnormalities the authors encountered during the processing procedures.

Thirdly, in Figure 9c, there seem to be a lot of features with a very high correlation (absolute value larger than 0.8), especially between ∆RMP and ∆APA. In this case, the SHAP value wouldn't make a lot of sense among these two data as they are so correlated. The reviewer would suggest the authors do some kind of dimension reduction analysis like PCA to strengthen their claim that "The main difference between Male and Female classifiers was in the ranking of ∆RMP and ∆APA." Also, the reviewer would expect that the authors wouldn't need all 14 features after PCA to achieve the same/better performance based on how correlated the features are in Figure 9c.

Minor Points

The citation style in Page 5 line 23 is inconsistent with the rest of the paper.

Reviewer #3: - The manuscript claims to introduce a novel pipeline, but the methodology heavily relies on a previously established multivariable analysis technique (cited in [47]),and is based on the hypothesis that sex is a critical factor in drug response, yet it fails to identify any new biomarkers beyond the 14 previously known.

- The classification task is restricted to a binary outcome (Class III vs. non-Class III drugs), which is overly simplistic for drug discovery applications. More nuanced classifications, such as drug efficacy and safety across multiple dimensions, would be more impactful and relevant.

- The use of six different machine learning models (Logistic Regression, SVM, Naive Bayes, XGBoost, Random Forest, K-Nearest Neighbor) represents an application rather than an innovation. These are standard models, and their application here does not constitute a novel methodological contribution. Anyone with access to these models could apply them to a similar dataset to produce predictions.

- The reported performance metrics (e.g., SVM accuracy of 0.8095 and specificity of 0.8217) can only tell the model is fitting well on the current features ( in distribution cross validation setting), and the binary classification task is not challenging to achieve scores like this, which do not demonstrate significant advancement in predictive capability.

- The reliance on SHAP for feature analysis is standard practice and does not yield novel insights. The identified important features are already well-documented in the literature, and the study does not provide new or surprising findings in this regard.

- Overall, the manuscript lacks sufficient innovation or impact to warrant publication in its current form.

**Have the authors made all data and (if applicable) computational code underlying the findings in their manuscript fully available?**

Reviewer #1: Yes

Reviewer #2: Yes

Reviewer #3: None

PLOS authors have the option to publish the peer review history of their article (what does this mean? ). If published, this will include your full peer review and any attached files.

**Do you want your identity to be public for this peer review?** For information about this choice, including consent withdrawal, please see our Privacy Policy .

Reviewer #1: No

Reviewer #2: No

Reviewer #3: No
---

## [Decision Letter · Decision Letter 1]

PCOMPBIOL-D-24-01299R1

A Novel Pipeline for Identifying Gender-Specific Class III Antiarrhythmic Drugs Based on In Vitro Measurements, In-Silico Models and Machine Learning Tools

PLOS Computational Biology

Dear Dr. Bai,

Thank you for submitting your manuscript to PLOS Computational Biology. After careful consideration, we feel that it has merit but does not fully meet PLOS Computational Biology's publication criteria as it currently stands. Therefore, we invite you to submit a revised version of the manuscript that addresses the points raised during the review process.

Please submit your revised manuscript within 60 days Feb 03 2025 11:59PM. If you will need more time than this to complete your revisions, please reply to this message or contact the journal office at ploscompbiol@plos.org. Please include the following items when submitting your revised manuscript:

We look forward to receiving your revised manuscript.

Kind regards,

Yang Lu, Ph.D.

Academic Editor

PLOS Computational Biology

Jason Haugh

Section Editor

PLOS Computational Biology

Feilim Mac Gabhann

Editor-in-Chief

PLOS Computational Biology

Jason Papin

Editor-in-Chief

PLOS Computational Biology

**Journal Requirements:**

1) Please ensure that the CRediT author contributions listed for every co-author are completed accurately and in full. At this stage, the following Authors require contributions: Xiaoshen Zhang. Please ensure that the full contributions of each author are acknowledged in the "Add/Edit/Remove Authors" section of our submission form. The list of CRediT author contributions may be found here: https://journals.plos.org/ploscompbiol/s/authorship#loc-author-contributions

2) We have noticed that you have uploaded Supporting Information files, but you have not included a list of legends. Please add a full list of legends for your Supporting Information files after the references list.

3) Please amend your detailed Financial Disclosure statement. This is published with the article. It must therefore be completed in full sentences and contain the exact wording you wish to be published. Please ensure that the funders and grant numbers match between the Financial Disclosure field and the Funding Information tab in your submission form. Note that the funders must be provided in the same order in both places as well.

- State the initials, alongside each funding source, of each author to receive each grant.

- State what role the funders took in the study. If the funders had no role in your study, please state: “The funders had no role in study design, data collection and analysis, decision to publish, or preparation of the manuscript.”

**Reviewers' comments:**

Reviewer's Responses to Questions

**Comments to the Authors:**

Reviewer #1: Thank you for incorporating my feedback, and for the clarifications. The addition of authors summary and further changes in the texts have clarified the significance and originality. The change of evaluation metrics also strengthens the scientific soundness.

Reviewer #2: Thank you for your detailed and thoughtful responses addressing my previous concerns. I have no further comments and am fully satisfied with the work.

Reviewer #3: Thank you for the responses and the revisions. After reviewing the responses, I still feel the manuscript falls short of addressing the major concerns.

1. The authors acknowledge that the methodology isn’t novel, and while I appreciate the focus on gender-specific drug responses, the approach is quite standard. Combining in vitro measurements, in silico simulations, and machine learning is useful, but it’s not particularly unique.

2. The issue of relying solely on previously known biomarkers still stands. While I understand that discovering new biomarkers wasn’t the goal, the study doesn’t offer any fresh insights about the ones used. Simply reiterating their relevance in a gender-specific context feels more like repackaging than advancing the field.

3. The binary classification task is very limiting. While I get that the authors wanted to focus on Class III and Class I drugs, this approach feels too narrow, especially when more nuanced classifications could have added real value. Acknowledging this as a limitation and suggesting future directions doesn’t solve the problem in the current study.

4. It also validates the point that the machine learning application here is more about routine analysis than a novel contribution.

5. The authors themselves confirm that this is more of a hypothetical analysis rather than methodological innovation. While SHAP and sensitivity analysis are fine tools, they’re standard, and the study doesn’t bring anything particularly new to the table.

**Have the authors made all data and (if applicable) computational code underlying the findings in their manuscript fully available?**

Reviewer #1: Yes

Reviewer #2: None

Reviewer #3: Yes

PLOS authors have the option to publish the peer review history of their article (what does this mean? ). If published, this will include your full peer review and any attached files.

**Do you want your identity to be public for this peer review?** For information about this choice, including consent withdrawal, please see our Privacy Policy .

Reviewer #1: No

Reviewer #2: No

Reviewer #3: No

**Figure resubmission:**
---

## [Decision Letter · Decision Letter 2]

PCOMPBIOL-D-24-01299R2

A Computational Pipeline for Identifying Gender-Specific Class III Antiarrhythmic Drugs Based on In Vitro Measurements, In-Silico Models and Machine Learning Tools

PLOS Computational Biology

Dear Dr. Bai,

Thank you for submitting your manuscript to PLOS Computational Biology. After careful consideration, we feel that it has merit but does not fully meet PLOS Computational Biology's publication criteria as it currently stands. Therefore, we invite you to submit a revised version of the manuscript that addresses the points raised during the review process.

Please submit your revised manuscript within 60 days Apr 05 2025 11:59PM. If you will need more time than this to complete your revisions, please reply to this message or contact the journal office at ploscompbiol@plos.org. Please include the following items when submitting your revised manuscript:

We look forward to receiving your revised manuscript.

Kind regards,

Yang Lu, Ph.D.

Academic Editor

PLOS Computational Biology

Jason Haugh

Section Editor

PLOS Computational Biology

**Journal Requirements:**

1) Please amend your detailed Financial Disclosure statement. This is published with the article. It must therefore be completed in full sentences and contain the exact wording you wish to be published.

2) Please ensure that the funders and grant numbers match between the Financial Disclosure field and the Funding Information tab in your submission form. Note that the funders must be provided in the same order in both places as well. Currently, the order of this grant "075-15-2022-304" is different in both places. In addition, this information "China Scholarship Council under Project No. 202206785002" is missing from the Funding Information tab.

Please indicate by return email the full and correct funding information for your study and confirm the order in which funding contributions should appear. 

**Reviewers' comments:**

Reviewer's Responses to Questions

Reviewer #1: The authors have addressed my concerns in the previous revision already. In the current revision, the authors are thus trying to clarifying the concerns the third reviewers have. The concerns are mainly about the originality and the limited task scope. The authors have revised the manuscript and title to no longer claiming the approach is novel but instead emphasize on its application to unique data, addressing a practically important problem and yielding interesting new findings in gender-specific context. Personally I have no further questions for the authors' responses.

Reviewer #2: Thank you for your responses. While the paper has certain limitations in terms of methodological innovation, the reviewer recognizes its value. The authors have generated a substantial amount of simulated data and conducted a rigorous analysis of biomarkers and the effects of class III drugs on atrial fibrillation (AF) patients in a gender-specific context. Additionally, the work highlights several intriguing directions for future research, including one with clinical potential for gender-based drug prescription. The results are sufficiently robust to warrant publication in their current form.

Reviewer #4: The paper presents a pipeline for identifying gender-specific Class III antiarrhythmic drugs using in vitro data, in silico models, and ML techniques. The sex-specific classifiers achieve high accuracy and provide interpretable biomarker insights, aligning with the goals of precision medicine. However, the study's novelty, particularly as a methods-focused paper, is insufficient and requires significant improvement.

1. I agree with the third reviewer that the paper's novelty is limited. PLOS Computational Biology emphasizes methodological innovation, and this study does not introduce a sufficiently novel method. The only notable aspect is the consideration of gender in classification. Expanding the analysis to include other influencing factors, such as age, weight, or gender-specific characteristics in other drugs or diseases, would make the study more impactful. Presenting only a pipeline is too limited.

2. Regarding the third reviewer’s suggestion for new biomarker discovery, the authors might not need to fully identify new biomarkers but could instead propose potential candidates using SHAP interpretability analysis. This would sufficiently address the requirement in a method-focused paper.

3. Given the lack of novelty, the authors should consider incorporating more nuanced classifications as proposed by third reviewer or collecting additional datasets.

4. If the paper aims to function as a pipeline, the authors should improve their GitHub repository by providing clear instructions and examples for users and publish a package which can make it transfer to other tasks, making it more accessible and practical for the research community.

**Have the authors made all data and (if applicable) computational code underlying the findings in their manuscript fully available?**

Reviewer #1: Yes

Reviewer #2: None

Reviewer #4: Yes

PLOS authors have the option to publish the peer review history of their article (what does this mean? ). If published, this will include your full peer review and any attached files.

**Do you want your identity to be public for this peer review?** For information about this choice, including consent withdrawal, please see our Privacy Policy .

Reviewer #1: No

Reviewer #2: No

Reviewer #4: No

**Figure resubmission:**

**Reproducibility:**

---

## [Editor Report · Decision Letter 3]

Dear Prof. Bai,

We are pleased to inform you that your manuscript 'Sex-specific Identification of Class III Antiarrhythmic Drugs Based on In Vitro Measurements, Computer models, and Machine Learning Tools' has been provisionally accepted for publication in PLOS Computational Biology.

Best regards,

Jason M. Haugh

Section Editor

PLOS Computational Biology

---

## [Editor Report · Acceptance letter]

PCOMPBIOL-D-24-01299R3

Digital Twin for Sex-specific Identification of Class III Antiarrhythmic Drugs Based on In Vitro Measurements, Computer models, and Machine Learning Tools

Dear Dr Bai,

I am pleased to inform you that your manuscript has been formally accepted for publication in PLOS Computational Biology. Your manuscript is now with our production department and you will be notified of the publication date in due course.

With kind regards,

Lilla Horvath
